# Spatially targeted inhibitory rhythms differentially affect neuronal integration

Drew B Headley[1]*[†], Benjamin Latimer[2][†], Adin Aberbach[2], Satish S Nair[2]*

[1]Center for Molecular and Behavioral Neuroscience, Rutgers University – Newark, Newark, United States; [2]Electrical Engineering and Computer Science, University of Missouri, Columbia, United States

## eLife Assessment

This **valuable** study assesses through simulations how several features of local cortical circuits - interneuron subtypes, their specific targeting of dendritic compartments, and certain brain rhythms - together affect the integration of synaptic inputs by a pyramidal cell. Employing several carefully considered simulation setups they **convincingly** demonstrate that beta rhythms are best suited to modulate and control dendritic Ca-spikes while gamma rhythms affect their coupling to somatic spiking, or how basal inputs are directly integrated into somatic spikes. However, the baseline setup may be idealized for the generation of the events in question and it would be beneficial if the similarity to the in-vivo activity regime was demonstrated further. The results will be relevant for neuroscientists studying local circuits or developing more abstract theories at the systems level.

*For correspondence:
dbh60@newark.rutgers.edu (DBH);
nairs@missouri.edu (SSN)

[†]These authors contributed equally to this work

Competing interest: The authors declare that no competing interests exist.

**Abstract** Pyramidal neurons form dense recurrently connected networks with multiple types of inhibitory interneurons. A major differentiator between interneuron subtypes is whether they synapse onto perisomatic or dendritic regions. They can also engender local inhibitory rhythms, beta (12–35 Hz) and gamma (40–80 Hz). The interaction between the rhythmicity of inhibition and its spatial targeting on the neuron may determine how it regulates neuronal integration. Thus, we sought to understand how rhythmic perisomatic and distal dendritic inhibition impacted integration in a layer 5 pyramidal neuron model with realistic dendrites supporting Na+, NMDA, and Ca2+ spikes. We found that inhibition regulated the coupling between dendritic spikes and action potentials in a location and rhythm-dependent manner. Perisomatic inhibition principally regulated action potential generation, while distal dendritic inhibition regulated the incidence of dendritic spikes and their temporal coupling with action potentials. Perisomatic inhibition was most effective when provided at gamma frequencies, while distal dendritic inhibition functioned best at beta. Moreover, beta modulated responsiveness to distal inputs in a phase-dependent manner, while gamma did so for proximal inputs. These results may provide a functional interpretation for the reported association of soma-targeting parvalbumin-positive interneurons with gamma and dendrite-targeting somatostatin interneurons with beta.

## Introduction

Cortical circuits are composed of connected networks of excitatory pyramidal neurons. To maintain a balanced level of excitability, they form dense reciprocal connections with local inhibitory interneurons. The two largest subclasses of interneurons in the cortex are parvalbumin positive (PV) and somatostatin positive (SOM) cells (*Tremblay et al., 2016*). A major distinction between PV and SOM interneurons is that they synapse onto different parts of the pyramidal neuron. PV interneurons tend to contact the soma and proximal dendrites (perisomatic), while SOMs target distal dendrites

(*Kawaguchi and Kubota, 1997*; *Wang et al., 2004*; *Kubota, 2014*; *Kubota et al., 2015*). Numerous studies have identified differences in how they regulate the responsiveness of local excitatory principal neurons (*Wilson et al., 2012*; *Lee et al., 2012*; *Atallah et al., 2012*). These differences could arise from their connectivity, intrinsic properties, or where they synapse onto pyramidal neurons. Indeed, the location of an inhibitory synapse qualitatively changes its effect on synaptic integration (*Kubota et al., 2015*; *Leleo and Segev, 2021*; *Jadi et al., 2012*; *Ferguson and Cardin, 2020*).

Synaptic integration in pyramidal neurons arises from the interplay between passive and active ion channels and dendritic morphology (*Leleo and Segev, 2021*; *Egger et al., 2020*; *Goetz et al., 2021*; *Ujfalussy et al., 2018*). Dendrites produce regenerative spiking events, namely $Na^+$, NMDA, and $Ca^{2+}$ spikes. Their initiation depends upon the spatiotemporal coordination of excitatory synapses and interaction with the depolarization of dendritic branches. Incorporating just NMDA spikes into a model neuron radically increases the complexity of its dendritic integration (*Beniaguev et al., 2021*). $Na^+$ and NMDA spikes can drive somatic spiking in an in vivo-like model neuron (*Goetz et al., 2021*). A $Ca^{2+}$ spike in the apical trunk converts a single somatic spike to a burst (*Leleo and Segev, 2021*; *Larkum et al., 1999*) and may allow pyramidal neurons to act as multi-stage integrators (*Poirazi et al., 2003*; *Larkum et al., 2009*). This complexity is only increased by including inhibition. The emission of bursts of action potentials is controlled by the timing and dendritic location of inhibition (*Leleo and Segev, 2021*).

The rhythmicity of inhibition at distal dendrites also affects somatic integration (*Li et al., 2013*). Reciprocal interactions between excitatory principal neurons and inhibitory interneurons give rise to rhythmic activities (pyramidal-interneuron network gamma or PING mechanism). To initiate the rhythm, excitatory principal neurons activate inhibitory interneurons that deliver feedback inhibition to principal neurons. This transiently suppresses principal neuron firing. As the inhibition wanes, principal neurons resume their activity and reengage the inhibitory population, starting a new oscillatory cycle. Some evidence suggests that different interneuron subtypes pace rhythms at different frequencies, with gamma oscillations depending on PV (*Sohal et al., 2009*; *Chen et al., 2017*; *Veit et al., 2017*) and beta oscillations on SOM (*Chen et al., 2017*). On the other hand, recent work has also found that beta/low-gamma rhythms in V1 engage both PV and SOM in their generation (*Onorato et al., 2025*; *Tahvili et al., 2025*).

Numerous cognitive processes are associated with these inhibitory rhythms in the cortex. Gamma oscillations are especially pronounced during stimulus presentation or behavioral initiation (*Headley and Weinberger, 2013*; *Murty et al., 2018*; *Muthukumaraswamy, 2010*). Beta rhythms occur during preparatory states or working memory (*Sanes and Donoghue, 1993*; *Lundqvist et al., 2016*). But fundamentally, these rhythms probably reflect different modes of local cortical activity and interregional communication (*Bastos et al., 2020*; *Bastos et al., 2015*; *Michalareas et al., 2016*). For instance, gamma oscillations are associated with feedforward transmission of information in cortical circuits, and beta oscillations may mediate feedback (*Bastos et al., 2020*; *Vezoli et al., 2021*), but see *Vinck et al., 2023*.

Given all this, it is important to better understand how the various integrative events in the dendritic tree are affected by the location and frequency of inhibition. Perhaps some rhythmic timescales are more effective in the dendrites, and others at the soma. Investigating this is beyond the reach of conventional experimental techniques. Moreover, computational models exploring how beta and gamma influence information processing have mostly relied upon simplified model neurons (e.g. *Tiesinga et al., 2004*; *Feng et al., 2019*). Thus, using a morphologically and biophysically detailed layer 5 (L5) pyramidal neuron model with $Na^+$, NMDA, and $Ca^{2+}$ spikes, we investigated how perisomatic and distal dendritic inhibition impacted dendritic integration and the modulation of these events by beta and gamma rhythmicity.

## Results

### Construction of a model cortical L5 pyramidal neuron

To study the effect of inhibitory rhythms on synaptic integration and somatic spiking, we adapted a previously published morphologically and biophysically detailed model of a cortical L5 pyramidal neuron (*Figure 1A*; see Methods for details). Modifications to the model were done in accordance with the published literature. In brief, this model featured a multicompartmental dendritic tree that

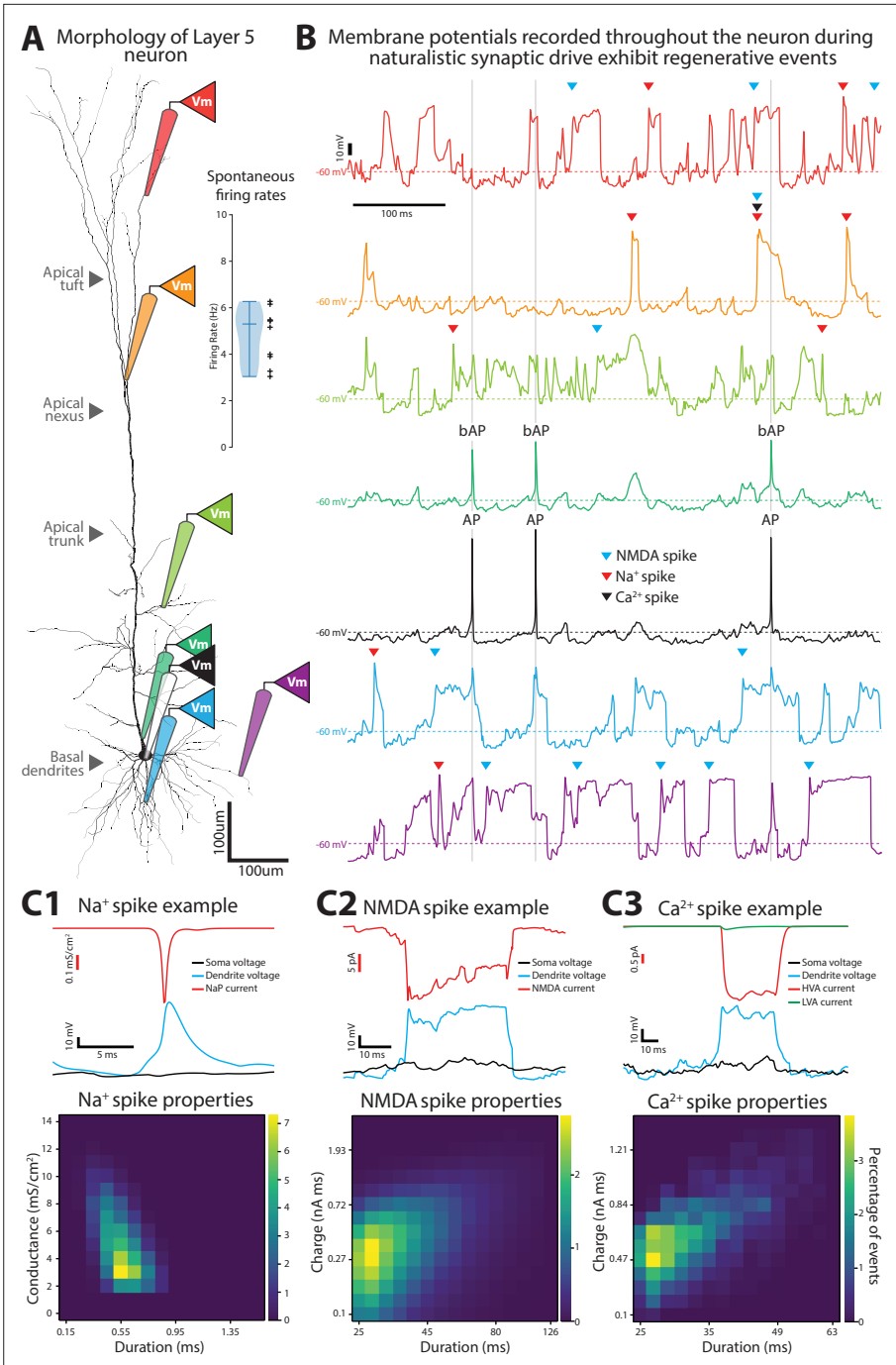

**Figure 1.** A model layer 5 pyramidal neuron with active dendrites. (**A**) The morphology of the neuron. Virtual recordings can be obtained from any desired compartment (colored pipettes). Inset, naturalistic presynaptic activity drives firing rates in our model like those in vivo. Each black cross is the mean rate for a different simulation. (**B**) Examples of membrane potentials recorded simultaneously across the dendritic tree (in color) and soma (black) during naturalistic drive. Regenerative events are indicated with arrows or text (AP: action potential, bAP: backpropagating action potential). (**C1–3**) Demonstration of our detection of dendritic spike events (top) and characterization of their properties (bottom). Events are binned according to the properties that were used in their detection. Bin edges for the event durations were not evenly set for panels **C2** and **C3**.

produced dendritic Na$^+$, NMDA, and Ca$^{2+}$ spikes, along with somatic action potentials that could backpropagate (*Figure 1B*). We distributed conductance-based synapses across the dendritic and somatic compartments with an average density of 2.16 excitatory and 0.22 inhibitory contacts per µm. PreCSIUYX synaptic drivers of excitatory synapses were drawn from a pool of 5200 point process sources that emulated correlated afferent drive with 2–8 synaptic contacts from the same presynaptic neuron. Inhibitory synapses were divided into two populations, those targeting the soma and proximal 100 µm of the dendrites (referred to as perisomatic) and those synapsing outside that area (referred to as distal). To capture excitatory/inhibitory (E/I) balance, a hallmark of cortical activity, the rate of inhibitory synaptic drive was a rescaled version of the rate of excitatory drive, lagged by 4 ms to emulate feedforward inhibition. Naturalistic presynaptic drive elicited a median firing rate of 5.3 Hz, in agreement with in vivo rates in cortex (*Figure 1A* inset, *Saiki et al., 2018*).

Dendrites were endowed with the following voltage-dependent conductances: a fast-inactivating Na$^+$ current ($I_{NaT}$), muscarinic K$^+$ current ($I_m$), fast non-inactivating K$^+$ current ($I_{Kv3.1}$), high voltage-activated Ca$^{2+}$ current ($I_{Ca\_HVA}$), low voltage-activated Ca$^{2+}$ current ($I_{Ca\_LVA}$), and Ca$^{2+}$ activated K$^+$ current ($I_{SK}$). As a result, the basal and apical dendrites could generate Na$^+$ and NMDA spikes (*Figure 1B*; *Goetz et al., 2021*). Dendritic Na$^+$ spikes were regenerative events lasting less than 1 ms that were not preceded by somatic action potentials (*Figure 1C1*; *Golding and Spruston, 1998*). An Na$^+$ spike was detected when the dendritic Na$^+$ channel conductance (gNa) was higher than 0.3 mS/cm$^2$, except when this threshold was reached within 5 ms after a somatic action potential, to distinguish from backpropagating action potentials.

NMDA spikes occur when adjacent NMDA-bearing synapses were synergistically recruited by a combination of glutamatergic activation and local depolarization (*Figure 1C2*; *Larkum et al., 2009*; *Schiller et al., 2000*). They typically lasted between 20 and 80 ms. They were detected when a compartment's membrane voltage exceeded –40 mV for at least 26 ms and NMDA current exceeded –100 pA ms of charge.

Ca$^{2+}$ spikes are depolarizations generated at the nexus of the apical trunk upon activation of voltage-gated Ca$^{2+}$ channels (*Figure 1C3*; *Schiller et al., 1997*; *Larkum and Zhu, 2002*). They lasted between 20 and 50 ms. To detect them, the membrane potential must exceed –40 mV for at least 26 ms, and the combined Ca$^{2+}$ currents (LVA and HVA) had to be 1.3 times higher than when the voltage criterion was reached ($t_v$). The Ca$^{2+}$ spike ended when this value fell to 1.15 times its value at $t_v$.

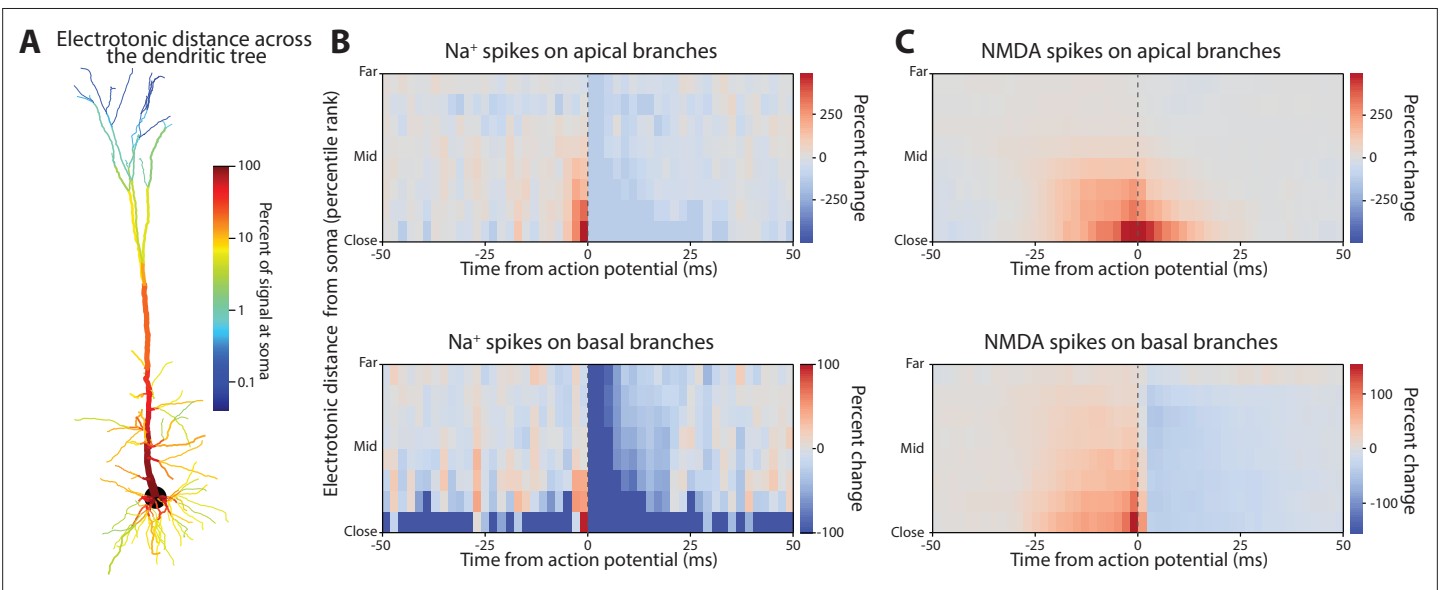

**Figure 2.** Influence of Na$^+$ and NMDA spikes on action potential generation. (**A**) Electrotonic distance between each dendritic compartment and the soma. (**B**) Dendritic compartments were grouped by their type (apical or basal) and electrotonic distance (percentile) from the soma. The percent change in Na$^+$ spike presence in those compartments relative to somatic spiking. Na$^+$ spikes increased immediately prior to action potentials in dendritic compartments that were electrotonically close to the soma. (**C**) Same format as **B**, but for NMDA spikes. These showed a similar degree of change, but a broader temporal coupling.

Altogether, under conditions that mirror in vivo afferent drive, our model reproduces the dendritic spikes of an L5 pyramidal neuron.

## Relationship between dendritic and somatic spikes

Dendritic spikes induced by synaptic activity are the principal drivers of somatic action potentials. Prior experimental and modeling work has found this to be the case for L2/3 and L5 pyramidal neurons (*Goetz et al., 2021*; *Larkum et al., 2009*; *Smith et al., 2013*; *Helmchen et al., 1999*; *Xu et al., 2012*). L5 pyramidal neurons have a substantially longer apical trunk, which increases the electrotonic distance of their apical tuft from the soma (*Figure 2A*) and diminishes the ability of tuft synapses to elicit action potentials. Voltage-gated $Ca^{2+}$ channels at the apical nexus compensate for this by producing a robust $Ca^{2+}$ spike that drives a burst of action potentials at the soma (*Poirazi et al., 2003*; *Larkum et al., 2009*). Underscoring that $Ca^{2+}$ spikes compensate for morphology, pyramidal neurons with shorter apical dendrites have weaker $Ca^{2+}$ spikes that only elicit a single spike (*Fletcher and Williams, 2019*; *Gidon et al., 2020*; *Larkum et al., 2007*). We thus assessed how dendritic spikes at different electronic distances from the soma were related to action potential generation.

Dendritic compartments differed in their degree of passive electrical coupling to the soma (i.e. electrotonic distance; *Figure 2A*). We measured this by injecting a 20 Hz sinusoidal current in each dendritic compartment and then calculating the ratio of the membrane voltage response at the soma over that at the dendrite. The apical trunk exhibited a relatively small attenuation ratio of ~10%. Progressing distally into the apical tuft, attenuation reached 0.1%. Basal dendrites showed greater attenuation than the apical trunk due to their smaller diameter. However, with an attenuation ratio reaching ~1%, their distal tips were still electrotonically closer than the apical tuft. Thus, large and long-lasting changes in membrane potential, like those produced by dendritic spikes, would be required to have any effect on the soma.

We examined this by measuring the spike-triggered average between somatic action potentials and the presence of dendritic spikes across the dendritic tree. To simplify the complex geometry of our model neuron, dendritic compartments were grouped into deciles by their electrotonic distance and whether they were on apical or basal branches. For each time lag from the somatic action potential, we measured the percent change in dendritic spike incidence from the mean rate across the entire simulation.

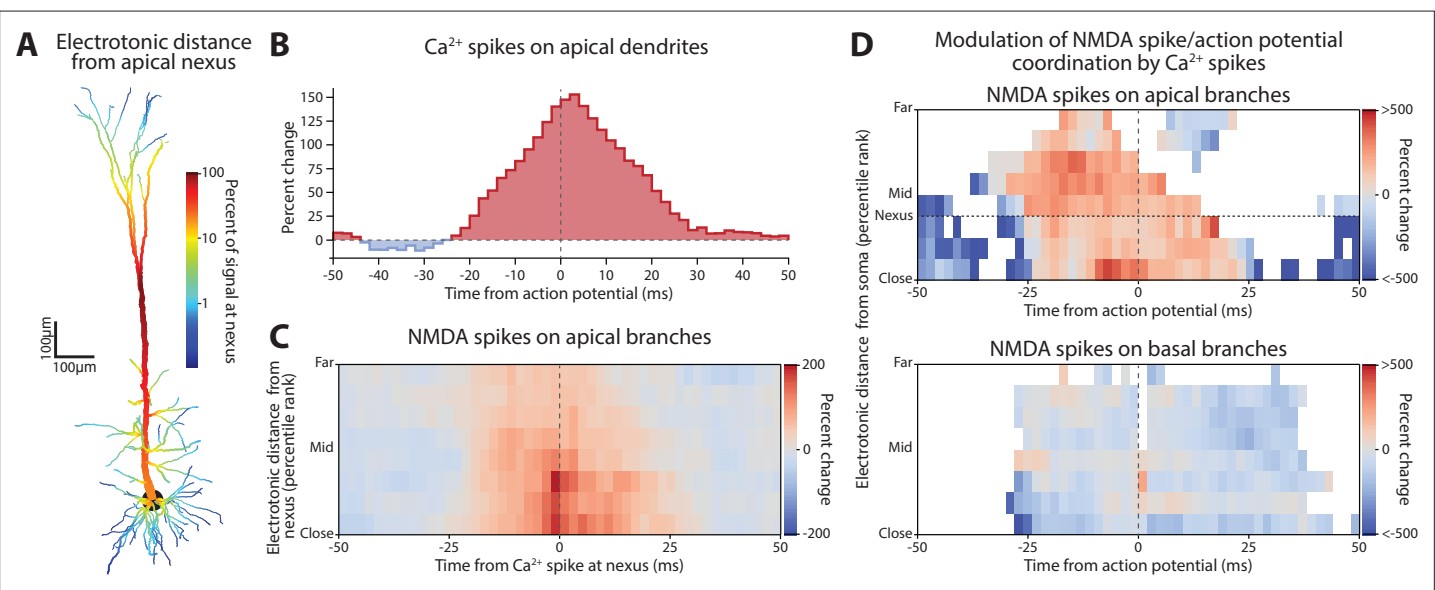

**Figure 3.** Influence of $Ca^{2+}$ spikes on action potential generation and their behavior as a second integrative mechanism. (**A**) Electrotonic distance between dendritic compartments and the apical nexus, where $Ca^{2+}$ spikes are generated. (**B**) Change in the incidence of $Ca^{2+}$ spikes at the nexus surrounding action potentials. (**C**) Percent change in NMDA spike presence in the apical dendrites centered on $Ca^{2+}$ spike initiation. (**D**) Percent change in NMDA spike coupling with action potentials during $Ca^{2+}$ spikes. Top, NMDA spikes in apical dendrites were more strongly coupled with action potentials during $Ca^{2+}$ spikes. Bottom, this was not the case for basal dendrites.

Dendritic Na$^+$ spikes increased 2–3 ms prior to somatic action potentials in both basal and apical dendrites (*Figure 2B*). This relationship was strongest for the compartments nearest the soma, with the rate of Na$^+$ spikes in the apical trunk increasing 300% over baseline prior to somatic action potentials and 100% in basal compartments. This relationship fell off as the dendritic spikes moved farther away from the soma, indicating that Na$^+$ spikes in distal branches had little direct influence on somatic spiking.

The incidence of NMDA spikes increased ~25 ms prior to somatic action potentials, much earlier than seen with dendritic Na$^+$ spikes (*Figure 2C*). But, like dendritic Na$^+$ spikes, NMDA spikes in the apical branch had a stronger coupling with somatic spiking than those in basal branches, and this effect dropped with distance from the soma. NMDA spikes in the apical branch persisted after an action potential, while those in basal dendrites did not, potentially because of the action potential after-hyperpolarization.

Ca$^{2+}$ spikes originate at the nexus, when the apical trunk first branches into the apical tuft. This region is electrotonically close to the entire apical trunk, facilitating the propagation of Ca$^{2+}$ spikes (*Figure 3A*). In our model, Ca$^{2+}$ spike occurrence increased within 20 ms of somatic action potentials (*Figure 3B*). Furthermore, we found that NMDA spikes in the apical dendrites tended to precede Ca$^{2+}$ spikes (*Figure 3C*). Our ionic current-based detection criteria distinguished between these phenomena despite their similar membrane voltage profiles. Since NMDA spikes in the apical tuft normally have a weak relationship to somatic spiking (*Figure 2C*), they may elicit somatic spiking indirectly by driving Ca$^{2+}$ spikes. Put another way, the apical nexus may serve as a thresholded nonlinearity for NMDA spikes in the apical tuft to drive action potentials (*Larkum et al., 2009*). To test this, we measured how a Ca$^{2+}$ spike changed the spike-triggered average between apical tuft NMDA spikes and action potentials (*Figure 3D*, top). This revealed that action potentials preceded by a Ca$^{2+}$ spike (by up to 20 ms) had increased coupling with apical NMDA spikes. No such change was seen in basal dendrites (*Figure 3D*, bottom).

## Effect of perisomatic and distal dendritic inhibition on controlling response gain

Subtypes of inhibitory interneurons synapse on distinct dendritic zones. PV interneurons mainly synapse perisomatically, while SOM interneurons target distal dendrites (*Kawaguchi and Kubota, 1997*; *Wang et al., 2004*; *Kubota, 2014*; *Kubota et al., 2015*). These differences may affect their modulation of synaptic integration (*Doiron et al., 2001*), either by shifting the threshold for evoking action potentials (a subtractive effect) or altering the slope of the relationship between excitation and firing rate (a divisive effect). There is some disagreement about the degree to which PV and SOM interneurons produce either of these effects (*Wilson et al., 2012*; *Lee et al., 2012*; *Atallah et al., 2012*), depending on the source of excitation and local circuitry (*Seybold et al., 2015*). So, before we applied beta and gamma rhythmic inhibition in our model, we studied the effect of tonically activating PV- and SOM-like inhibitory synapses.

We doubled the rate of the inhibitory presynaptic drive onto either the PV-targeted perisomatic compartments (somatic and dendritic compartments within 100 µm) or the SOM-targeted distal (>100 µm) dendritic branches. Both decreased the firing rate of the pyramidal cell from 5.5 Hz to less than 1 Hz (*Figure 4A*; control: 5.5±0.85 Hz; distal: 0.20±0.15 Hz; perisomatic: 0.70±0.31 Hz; mean ± SD). These comparable changes may reflect either subtractive or divisive effects and could derive from different mechanisms.

To isolate these factors, we first assessed how doubling inhibition affected action potential initiation at the soma. A series of current pulses were injected into the soma to measure the relationship between firing rate and injected current (f-I curve), which captures the gain function of the neuron (*Figure 4B*). This revealed that both perisomatic and dendritic inhibition shifted the current threshold for action potential initiation. Such an effect is subtractive. In addition, perisomatic inhibition decreased the slope of the f-I relationship compared with the control, which is consistent with a divisive effect.

Although perisomatic inhibition produced the strongest subtractive effect, distal dendritic inhibition reduced firing rate the most (*Figure 4A*). How can we reconcile these discordant findings? One possibility is that distal inhibition substantially reduces Na$^+$, NMDA, or Ca$^{2+}$ spikes. To determine this, we examined the overall rate of dendritic spike events. Perisomatic inhibition did not affect dendritic events compared to the control condition (*Figure 4C–E*). By contrast, dendritic inhibition decreased

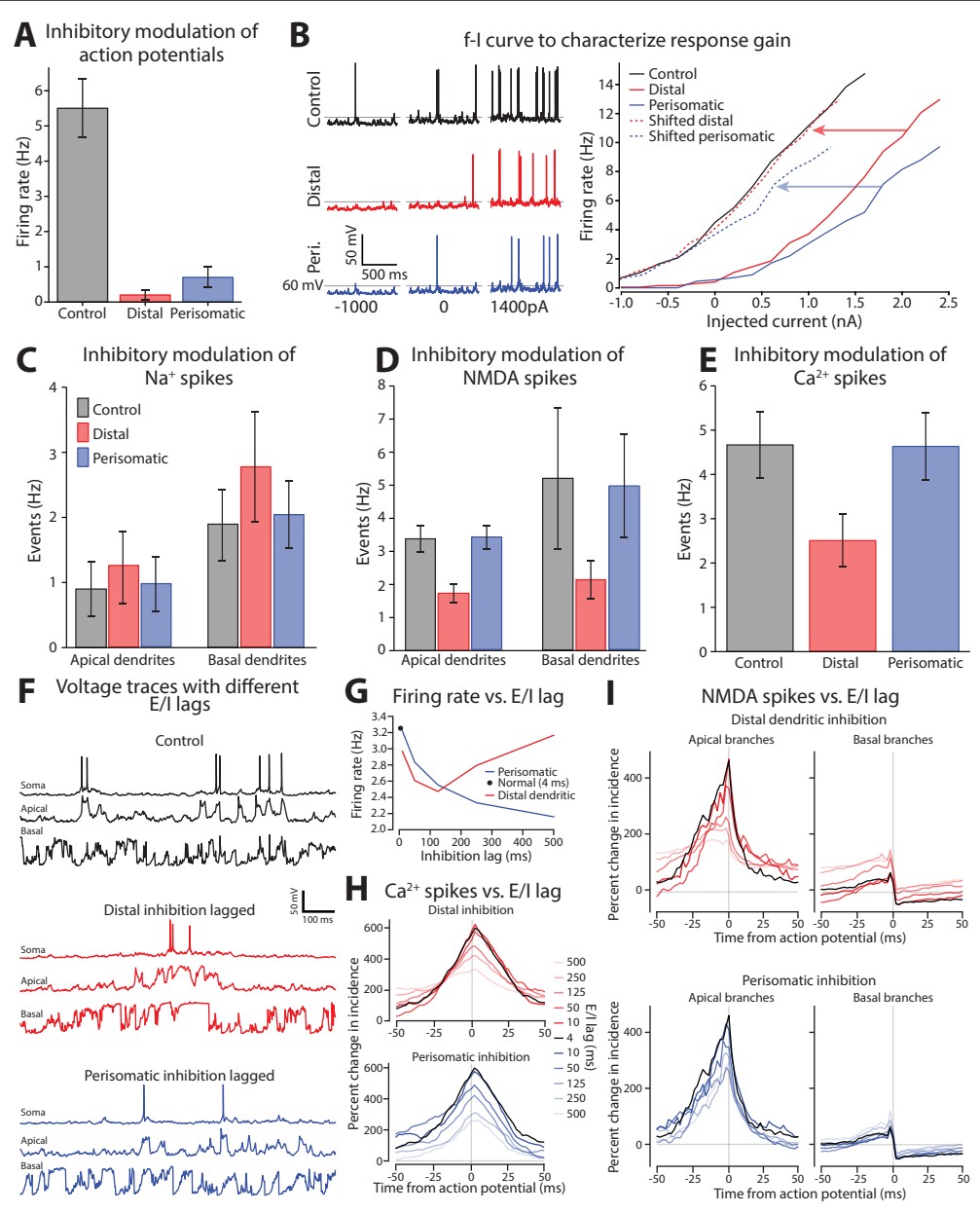

**Figure 4.** Distal dendritic and perisomatic inhibition reduce action potential generation through different mechanisms. (**A**) Action potential rate during periods with normal inhibitory tone (control), double rate on distal branches, or double rate on perisomatic. Both increases in inhibition dramatically reduced the firing of somatic action potentials. (**B**) Somatic excitability was measured by delivering current steps during the control, distal, and perisomatic inhibition states. Left, example somatic voltage responses to current steps. Right, spike frequency versus current (f–I) curve for each condition. The threshold for evoking an action potential shifted with 2× distal and perisomatic inhibition. Perisomatic, but not dendritic, inhibition changed the f-I slope (compare dashed lines with solid black). (**C**) Impact of altered dendritic inhibition on rate of Na+ spikes in apical and basal dendrites. (**D**) Same format as **C**, but for NMDA spikes. (**E**) Rate of Ca2+ spikes in the apical dendrites. Basal dendrites lacked Ca2+ spikes and were excluded. All error bars are mean ± standard deviation. (**F**) Examples of membrane potential recorded in control (top), and both distal (middle) and proximal (bottom) inhibition lagged by 500 ms. (**G**) Change in firing rate for control (black dot) and for perisomatic (blue) and distal (red) lags in inhibition from 0 to 500 ms. (**H**) Change in incidence of Ca2+ spikes for distal (red, top) and proximal (blue, bottom) inhibition. The control case is shown in black in both panels. (**I**) Same as (**H**) but for NMDA spikes.

NMDA and $Ca^{2+}$ spikes (*Figure 4D and E*). $Na^+$ spikes were relatively unaffected (*Figure 4C*). This lack of effect may arise from a shortening of the inactivation for voltage-gated $Na^+$ channels balancing out the loss of excitatory drive.

## Distinct excitation/inhibition balance effects of perisomatic and distal dendritic inhibition

The previous analysis found distinct effects on neuronal gain and dendritic spiking from tonic changes in perisomatic and distal dendritic inhibition. But in vivo inhibition is dynamic and time-varying. PV and SOM interneurons form dense interconnections with local pyramidal neurons, supplying time-lagged feedback inhibition. This helps maintain the balance of excitation and inhibition (E/I) in the network by ensuring that an overall increase in excitatory activity is rapidly counterbalanced by proportionate inhibition.

To emulate situations where E/I balance is important, we increased the dynamic variation in excitatory drive (see Methods). As with the previous simulations, the rate of inhibitory synaptic drive was a lagged and rescaled version of the overall excitation rate. Normally, that lag is 4 ms, in line with experimental estimates (*Wehr and Zador, 2003*). To probe whether perisomatic or distal dendritic inhibition has distinct effects on E/I balance, we independently varied their lags (*Figure 4F–I*). One was kept at the nominal 4 ms lag, while the other was extended. Increasing either of their lags by 500 ms produced obvious differences in the emission of dendritic spikes and their coordination with action potentials (*Figure 4F*). Normally, dendritic spikes are evenly distributed in time and drive somatic action potentials. Lagging distal dendritic inhibition clustered dendritic spikes together in time, during which action potentials were emitted. Lagging perisomatic inhibition did not affect the spacing of dendritic spikes but decreased their coupling with somatic spiking.

We systematically characterized these lag effects for the following spiking events modulated by tonic changes in inhibition: action potentials, $Ca^{2+}$, and NMDA spikes (*Figure 4G*). Increasing the lag of perisomatic inhibition lowered action potential firing, while for distal dendritic inhibition, the firing rate decreased out to a lag of 125 ms and then returned to normal at 500 ms. To better understand these effects, we calculated the cross-correlation (CC) between dendritic spikes and action potentials. Increasing the lag decreased the coordination between $Ca^{2+}$ and somatic spikes (*Figure 4H*). For distal dendritic inhibition, this decrease was accompanied by a broadening of their temporal relationship, while a sharp temporal relationship was maintained in the perisomatic case. A similar pattern was observed with NMDA spikes on apical branches (*Figure 4I*). Basal branches, on the other hand, were relatively unaffected.

In summary, distal and perisomatic inhibition modulate the coupling between synaptic drive and spiking through distinct processes. And while for both cases, the normal 4 ms E/I lag produced maximal conversion of dendritic spiking events into action potentials, extending this lag exerted distinct effects.

## Effect of beta and gamma rhythmic inhibition on neuronal integration

The mechanisms modulating neuronal responsiveness during tonic inhibition of somatic and dendritic compartments may extend to rhythmic inhibition. Thus, we emulated beta and gamma rhythmic input (*Figure 5A and F*). Depths of modulation were set to similarly entrain action potentials (*Figure 5B and G*) and were comparable to spontaneous and optogenetically induced gamma and beta bursts seen in vivo (*Amir et al., 2018*; *Onorato et al., 2020*; *Adesnik, 2018*; *Murthy and Fetz, 1992*). Beta rhythmic inhibition was modeled as a 16 Hz sinusoidally modulated rate (20% depth) of the Poisson processes driving inhibitory synapses. Gamma rhythmic inhibition was a 64 Hz sinusoidal modulation (40%) of inhibitory synapses. For both cases, excitatory synaptic drive was a stable Poisson process.

Initially, we delivered beta rhythmic inhibition to the distal dendritic compartments and gamma to the perisomatic. Even though both rhythms produced similar depths of modulation of somatic action potentials, the underlying causes were distinct. The phase of beta modulated the occurrence of $Ca^{2+}$, NMDA, and $Na^+$ spikes, with each showing an ~75% depth of modulation with respect to their mean level (*Figure 5C–E*). These changes were seen across the entire dendritic tree, spanning sites electrotonically close to and far from the soma. In addition, the impact on $Na^+$ spikes was unexpected (*Figure 5E*), since delivery of the same inhibition tonically had little effect. By contrast, gamma had

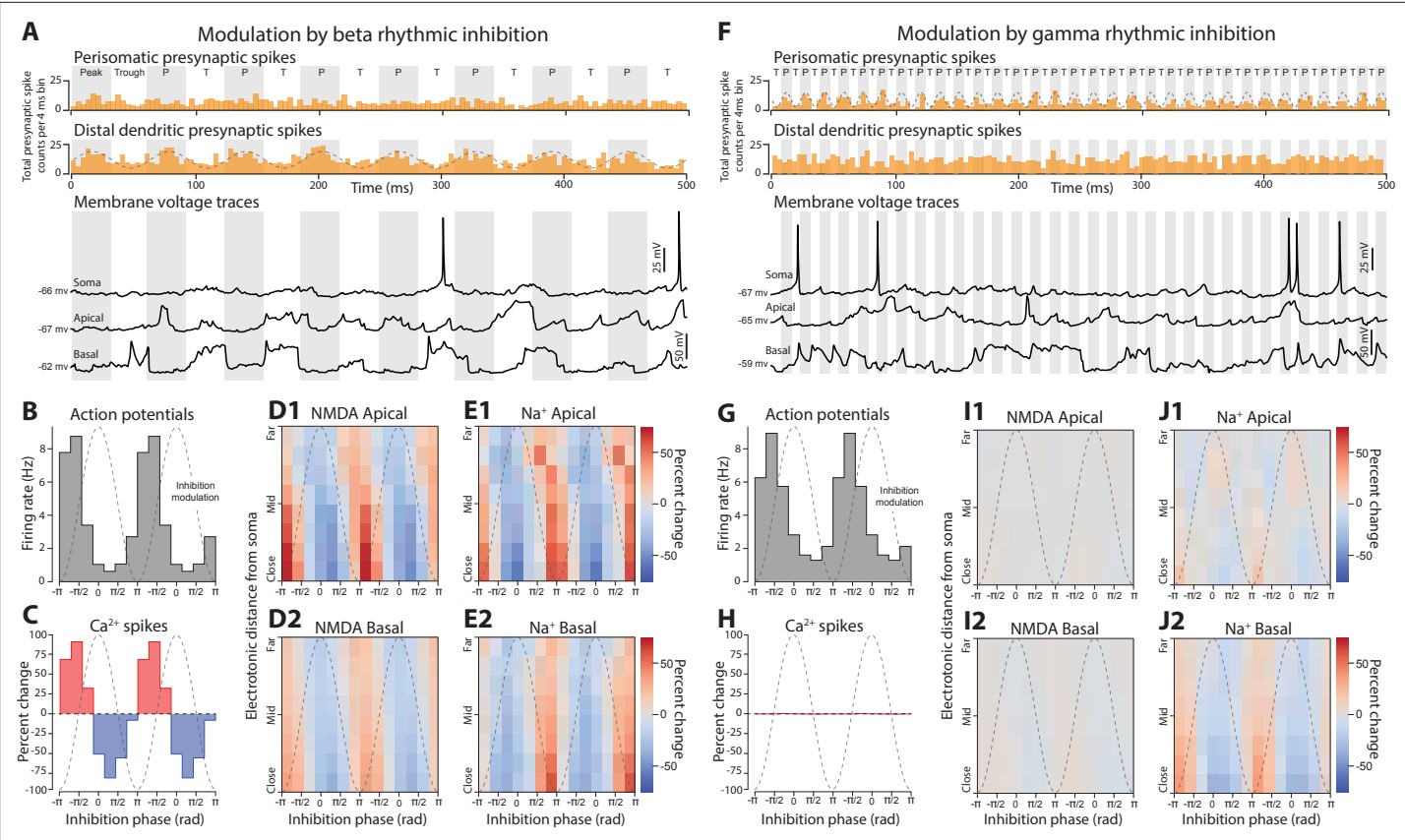

**Figure 5.** Phase-dependent effects of beta and gamma rhythmic inhibition on dendritic spikes. (**A**) Example data from the beta rhythmic inhibition simulation. Top, presynaptic spike counts. Bottom, voltage traces from somatic and dendritic compartments. Grayed periods are when inhibitory presynaptic spikes are peaking. (**B**) Action potential rate as a function of the phase of the beta rhythm. Dashed gray line shows the modulation of inhibitory drive with respect to phase. (**C**) Percent change in $Ca^{2+}$ spike presence at apical nexus by beta phase. (**D1–2**) Percent change of NMDA spike presence in apical (1) and basal (2) dendrites stratified by electronic distance from the soma. (**E1–2**) Same as **C**, but for $Na^+$ spikes. (**F, G, H, I1–2, J1–2**) Same format as above, but with events binned by the phase of gamma rhythmic inhibition. For all graphs, phase is given in radians with inhibition at a minimum for $-\pi$ and maximum at 0.

The online version of this article includes the following figure supplement(s) for figure 5:

**Figure supplement 1.** Phase-dependent effects on dendritic spikes of beta and gamma rhythmic inhibition delivered to opposite areas of the neuron.

virtually no effect on dendritic spikes (***Figure 5F–H***). Its strongest impact was on $Na^+$ spikes in basal dendritic compartments that were electronically close to the soma.

To uncover the modulation of action potentials by gamma, we turned our attention to somatic action potential initiation (***Figure 6***). For these analyses, we divided the inhibitory rhythm into two phases. During the *peak* phase, inhibition was greater than its mean rate, while during the *trough* phase, inhibition was lower (see ***Figure 5A and F***). We found that the somatic action potential voltage threshold shifted lower during the 'trough' phase of gamma, when inhibition was at its weakest (***Figure 6A1***) and without any change in the mean membrane voltage (***Figure 6A2***). This is consistent with gamma phase modulating the shunting of voltage-gated $Na^+$ channel currents, which occurs when GABAergic synapses co-locate with the channels mediating action potential initiation (***Rojas et al., 2011***). We also observed changes in action potential initiation to beta rhythmic inhibition, but through a different mechanism. During the 'peak' phase of beta, when inhibition was maximal, the threshold for evoking an action potential increased, which may reflect an 'off-path' shunting of excitatory current away from the soma and toward the dendrites (***Figure 6B1***; ***Gidon and Segev, 2012***). Additionally, there was a decrease in membrane voltage during the peak phase, which may correspond to decreased excitation arising from the suppression of dendritic spikes (***Figure 6B2***).

Accompanying these effects were changes in the rate of dendritic spikes compared with the Poisson inhibition case. Beta increased the rate of $Na^+$ (+16.6%) and $Ca^{2+}$ spikes (+15.1%) but decreased the

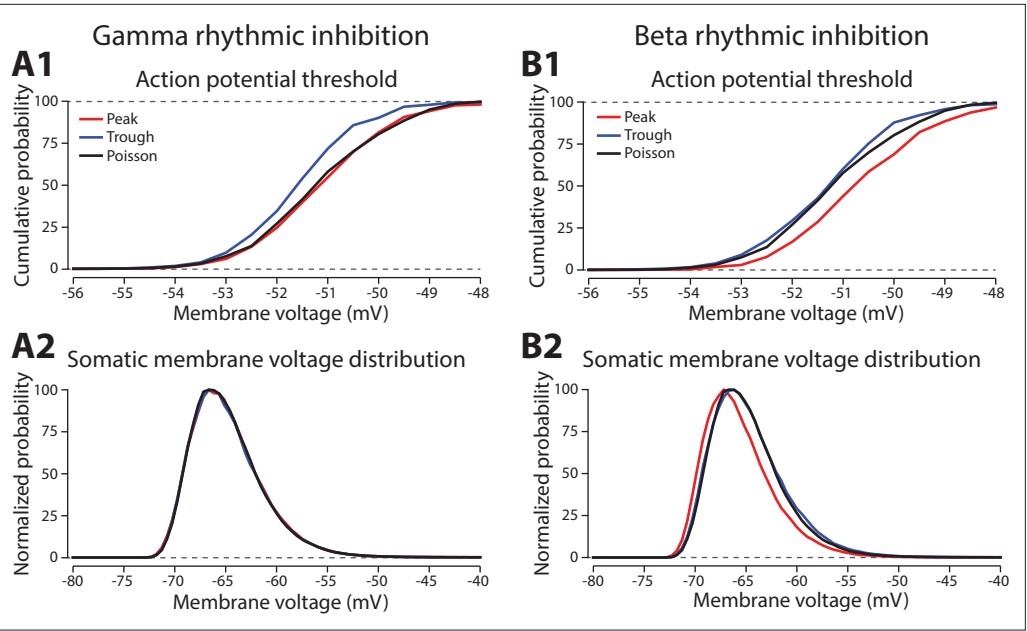

**Figure 6.** Phase-dependent effects of gamma and beta rhythmic inhibition on somatic excitability. (**A1**) A cumulative probability plot of the distribution of somatic membrane potentials 1 ms prior to an action potential, sorted by whether they occurred during the gamma phase with maximal (peak, red line) or minimal (trough, blue line) inhibitory drive. Poisson (black) had no rhythmic modulation, but the same mean inhibitory rate. (**A2**) Probability distribution of somatic membrane voltage as a function of gamma phase, normalized to the peak probability value. Lines have the same color scheme as in **A1**. (**B1**) Same format as **A1**, but for the beta rhythm. (**B2**) Same format as **A2**, but for the beta rhythm.

The online version of this article includes the following figure supplement(s) for figure 6:

**Figure supplement 1.** Phase-dependent effects on somatic excitability of beta and gamma rhythmic inhibition delivered to opposite areas of the neuron.

rate of NMDA spikes (–10.7%). Gamma caused no change in NMDA (+0.2%) and a weak increase in Na$^+$ spikes (+6.4%), but a robust gain in Ca$^{2+}$ spikes (+37.9%).

We next switched the locations on the pyramidal neuron targeted by the beta and gamma rhythms to disassociate the frequency of inhibition from its location. Beta rhythmic inhibition was delivered perisomatically and gamma rhythmic inhibition to distal dendrites. While phase modulation of firing rate was maintained with both rhythms, the overall level of spiking was dramatically reduced (*Figure 5—figure supplement 1A and E*). Neither rhythm modulated Ca$^{2+}$ or NMDA spikes (*Figure 5—figure supplement 1B, C, F, and G*). It is likely that the slow timescale of Ca$^{2+}$ and NMDA spikes, ~50 ms, is not optimal for the fast periodicity of the gamma rhythm, which cycles every ~15 ms. In agreement with this, Na$^+$ spikes, which last less than 1 ms, did show modulation by gamma rhythms delivered to the distal dendrites (*Figure 5—figure supplement 1D and H*).

Swapping the location of beta and gamma synapses altered their effects on somatic excitability. Gamma rhythmic inhibition on the dendrites had minimal or no impact on action potential threshold, but did shift the somatic membrane potential more negative (*Figure 6—figure supplement 1A*). This hyperpolarization was not dependent on gamma phase and likely reflected an overall decrease in the rate of dendritic spikes that could supply excitatory drive to the soma (NMDA: –26.1%, Na$^+$: –12.2%, and Ca$^{2+}$: –39.9% compared with the Poisson inhibition case). By contrast, delivering beta rhythmic inhibition to the soma raised the action potential threshold and hyperpolarized the membrane potential during the peak phase (*Figure 6—figure supplement 1B*). It also reduced the incidence of dendritic spikes (NMDA: –17.1%, Na$^+$: –11.5%, and Ca$^{2+}$: –7.6%).

Putting all this together, the effectiveness of rhythmic inhibition depends on where it impinges upon the neuron. Beta rhythms targeting the distal dendrites modulate the incidence of dendritic spikes in a phase-dependent manner, while gamma rhythms delivered perisomatically phase-modulate

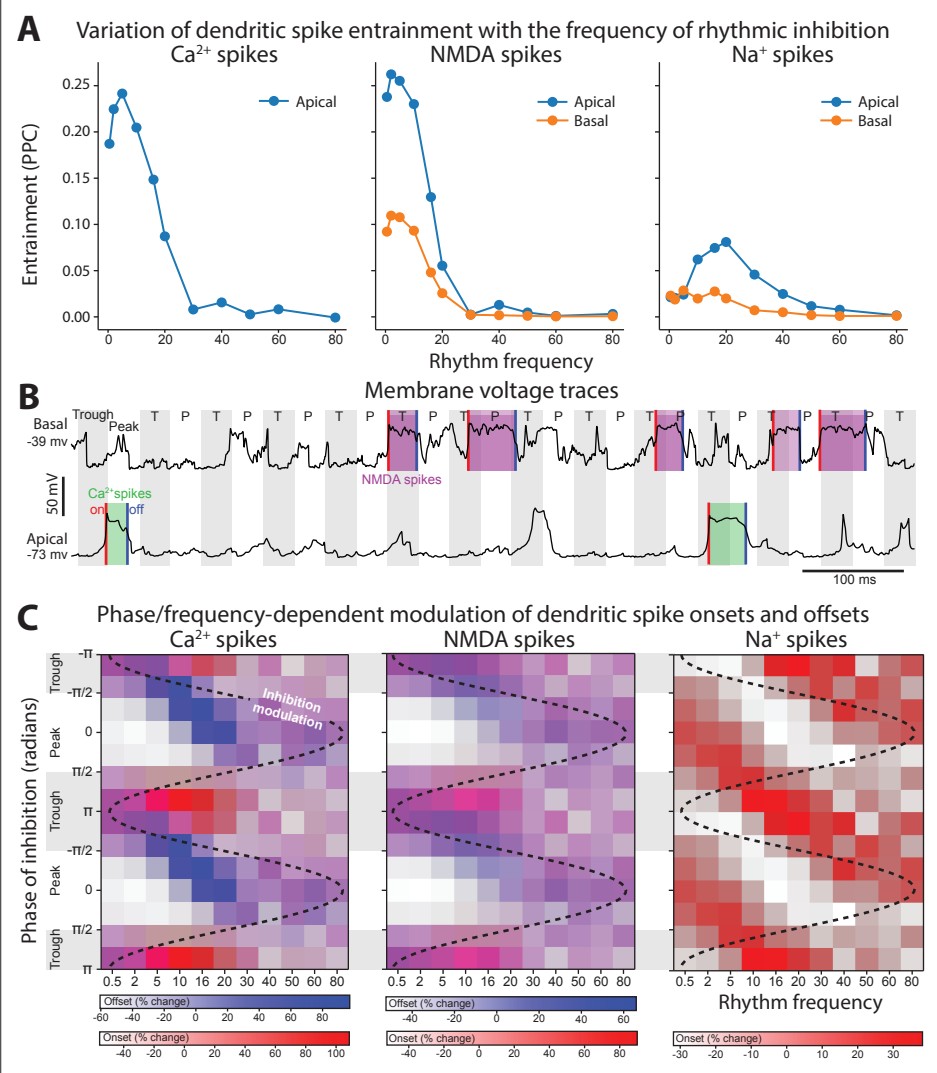

**Figure 7.** Frequency- and phase-dependent effects of inhibitory rhythms on the distal dendrites. (**A**) Entrainment to an inhibitory rhythm delivered to the distal dendrites varied with its frequency. Higher frequencies were less able to entrain dendritic spikes. Entrainment tended to be strongest for apical (blue) over basal (orange) compartments. (**B**) Example voltage traces from dendritic compartments in either the distal basal or apical branches. Gray shading denotes the period where the inhibitory rhythm troughs occurred. For the basal segment, NMDA spikes were shaded in purple, while in the apical segment, $Ca^{2+}$ spikes were shaded in green. Dendritic spike onsets denote with red lines, and offsets with blue lines. (**C**) Percent change from the mean in the rate of dendritic spike onsets (red gradient) and offsets (blue gradient) as a function of rhythm frequency and phase. Purple regions denote phase/frequency combinations where both onsets and offsets were elevated, while regions with either just blue or red indicate that offsets or onsets preferentially occurred, respectively. We did not determine an offset for $Na^+$ spikes due to their transience (~1 ms).

somatic excitability. Swapping the locations of these rhythms diminishes these effects and lowers overall excitability.

## Frequency-specific effects of rhythmic inhibition on neuronal integration

Having demonstrated that the location of beta and gamma rhythmic inhibition impacts their effectiveness, we next determined its frequency specificity. To do this, we varied the frequency of rhythmic inhibition between 0.5 and 80 Hz on either the perisomatic or distal dendritic neuronal compartments. Starting with distal dendrites, increasing inhibition frequency above 20 Hz diminished its entrainment

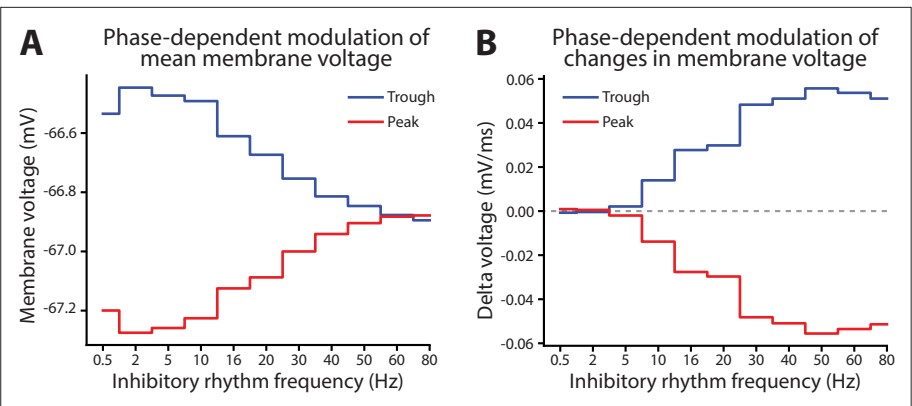

**Figure 8.** Frequency- and phase-dependent effects of inhibitory rhythms on the perisomatic region. (**A**) The mean somatic membrane potential during either the trough or peak phase of the inhibitory rhythm. (**B**) Mean of the distribution of somatic membrane potential fluctuations as a function rhythm phase and frequency. Fluctuations were measured across the entire simulation time as the difference in membrane potential at 1 ms delays. For both graphs, red lines are peaks and blue lines are troughs.

of NMDA, Na$^+$, and Ca$^{2+}$ spike onsets (*Figure 7A*). The falloff in entrainment was most pronounced above 20 Hz. Curiously, Na$^+$ spikes exhibited a preferential entrainment at 20 Hz. In general, entrainment was strongest in the apical dendrites.

The previous analysis considered the entrainment of dendritic spike *onsets*, but NMDA and Ca$^{2+}$ spikes also exhibit *offsets* that could also be modulated by rhythmic inhibition. Indeed, examination of voltage traces in the dendrites during beta rhythmic inhibition revealed that NMDA and Ca$^{2+}$ spike onsets tended to occur during the trough, while offsets happened during the peaks (*Figure 7B*). To quantify this, we plotted the percent change in the probability of dendritic spike onsets and offsets with respect to both the phase and frequency of the inhibitory rhythm (*Figure 7C*). For frequencies less than 5 Hz, there was minimal phase separation between the onsets and offsets of Ca$^{2+}$ or NMDA spikes; both events occurred near the rhythm trough, when inhibition was at its weakest. As frequency increased up to 20 Hz, the preferred phase of dendritic spike offsets migrated toward the peak phase, where inhibition is strongest. This phase separation effect was strongest for Ca$^{2+}$ spikes. Thus, beta band frequencies exhibit unique coordination with dendritic spikes: they are the fastest rhythm capable of entraining them and align with their initiation and cessation in a phase-dependent manner.

Turning to perisomatic inhibition, we again varied the frequency of the inhibitory rhythm. Lower frequency inhibition produced phase-dependent shifts in the mean membrane potential (*Figure 8A*). The trough of inhibition depolarized the soma, while the peak of inhibition had the opposite effect. As the frequency increased, this phase-dependent difference went away, vanishing above 50 Hz. By contrast, as frequency increased, the bias in momentary changes in the membrane potential diverged between peaks and troughs (*Figure 8B*). During the peak phase, membrane voltage fluctuations were biased negative, while during the trough, they were biased positive. This effect increased with frequency, peaking at 50 Hz and then declining modestly. Together, these effects make gamma frequencies unique in keeping the mean membrane potential equivalent between phases but biasing its fluctuations toward depolarizing or hyperpolarizing with phase. During the trough, there was an excess of depolarizing membrane potential fluctuations. Since the rate of excitatory synaptic drive was independent of phase, this suggests that its responsiveness to excitatory inputs increased in a phase-dependent manner with gamma.

## Modulation of dendritic spikes during oscillatory bursts

In vivo, beta and gamma rhythms occur as bursts lasting less than a few hundred milliseconds. Since the previous analyses relied on tonically delivered rhythms, we verified that similar effects were observed with oscillatory bursts. Gamma and beta bursts were delivered to the same model with mean depth of modulation like the tonic case (*Figure 9A and F*; see Methods for details).

Under the burst regime, both rhythms mirrored their behavioral effects. Gamma bursts entrained spiking, with entrainment strongest during the middle of the burst (*Figure 9B*). As with the tonically

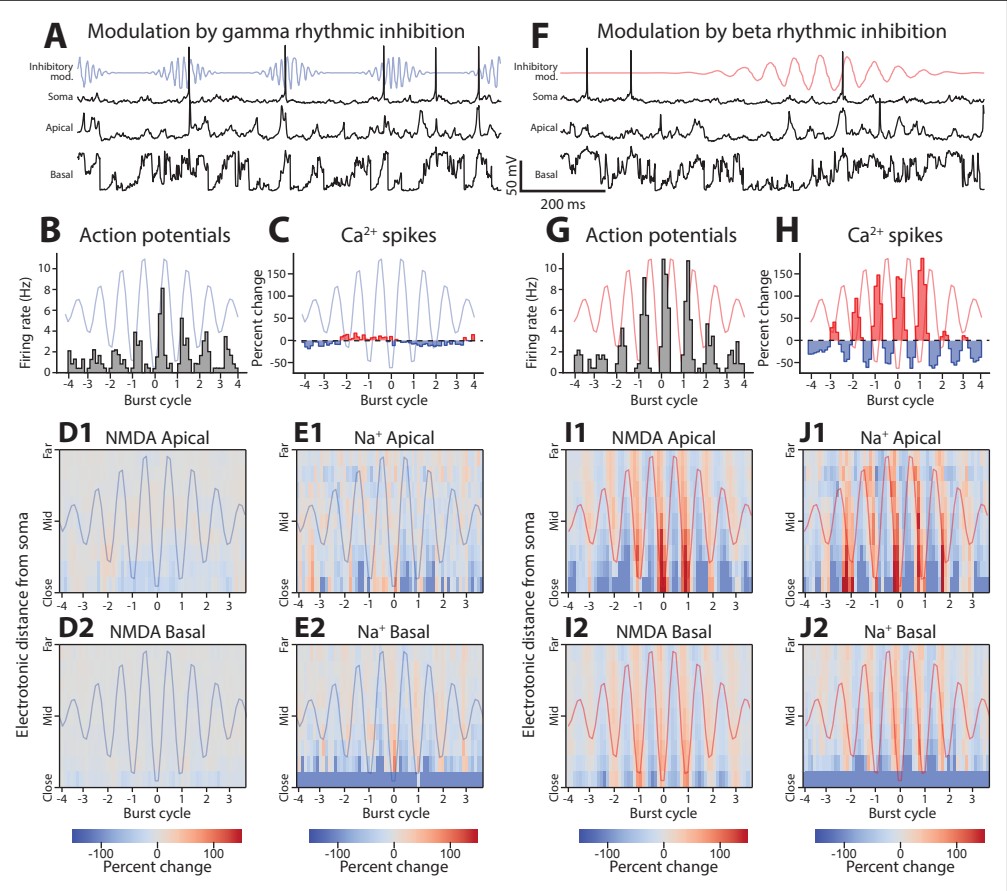

**Figure 9.** Phase-dependent effects of gamma and beta bursts on dendritic spikes. (**A**) Example data from the gamma rhythmic inhibition simulation. Top, somatic potential (black line), with firing rate of perisomatic inhibitory synapses (blue line). Middle, voltage trace of apical compartment. Bottom, voltage trace of basal compartment. (**B**) Action potential rate as a function of the phase of the gamma rhythm. Blue line shows the modulation of inhibitory drive with respect to phase. (**C**) Percent change in Ca²⁺ spike presence at apical nexus by gamma phase. (**D1–2**) Percent change of NMDA spike presence in apical (1) and basal (2) dendrites stratified by electronic distance from the soma. (**E1–2**) Same as **C**, but for Na⁺ spikes. (**F, G, H, I1–2, J1–2**) Same format as above, but with events binned by the phase of beta rhythmic inhibition. For all graphs, cycle number is given relative to the amplitude peak of the burst.

imposed rhythm, there was none or minimal modulation of Ca²⁺ (*Figure 9C*), NMDA (*Figure 9D*), and Na⁺ spikes (*Figure 9E*). Beta rhythms entrained somatic action potentials (*Figure 9G*), Ca²⁺ spikes (*Figure 9H*), NMDA (*Figure 9I*), and Na⁺ spikes (*Figure 9J*). These modulations were evident within the first few cycles of a burst, suggesting that they did not require a buildup or evolving entrainment of an underlying process.

## Effect of beta and gamma rhythms on responding to clustered synaptic drive

The results so far suggest that beta and gamma rhythms modulate synaptic integration through different mechanisms depending on their phase and the location of the synapses on the dendritic tree. To examine this further, we added patches of concentrated excitatory synaptic inputs onto either the distal or proximal dendrites (*Figure 10A*), with densities similar to functional clusters in vivo (*Iacaruso et al., 2017*; *Fu et al., 2012*). Coactivated inputs were simulated by driving each synapse with a jittered (2 ms) Poisson process. We ran six separate simulations, either with the distal or proximal clusters engaged, and under conditions of Poisson, beta, or gamma rhythmic inhibition.

To capture how beta and gamma influence synaptic integration, we measured the CC between presynaptic activations at the clustered input and somatic spiking. Separate cross-correlograms were

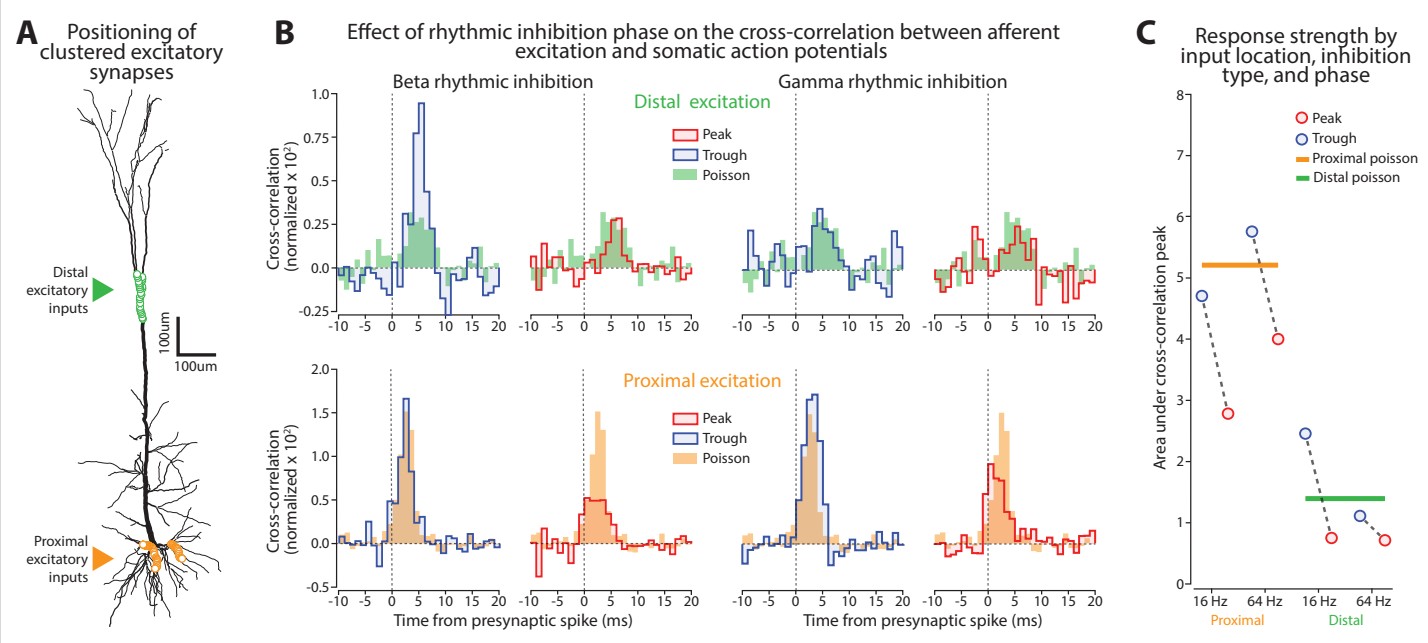

**Figure 10.** Effect of beta and gamma rhythms on responsiveness to synaptic inputs targeting distinct regions of the dendritic tree. (**A**) Schematic of the location for clustered excitatory synaptic inputs. (**B**) Normalized cross-correlation between synaptic drive onto a clustered input and spiking at the soma, stratified by whether the presynaptic spike arrived during the peak (red line) or trough (blue line) of the rhythm. Solid bars correspond to the Poisson stimulation case where inhibition was not rhythmically modulated. Top left, effect of beta on distal inputs. Top right, effect of gamma on distal inputs. Bottom left, effect of beta on proximal inputs. Bottom right, effect of gamma on proximal inputs. (**C**) Summary of effects in panel **B** where the strength of each normalized cross-correlation was measured as its area under the curve. Dots are connected by dashed gray lines if the data points came from the same simulation but at different phases of the rhythm. Solid horizontal lines reflect the cross-correlation strength in the Poisson inhibitory case (no rhythmicity).

calculated depending on whether the presynaptic spikes occurred during the peak or trough phase of the inhibitory rhythm. This necessarily introduced spurious periodicities into the cross-correlogram that were compensated (see Methods for details). Relative to the arhythmic Poisson inhibition case, beta rhythms enhanced the transmission of distal inputs when inhibition was low (trough phase) and suppressed them when inhibition was high (peak phase, *Figure 10B*, top left). Proximal inputs were either unaffected or moderately suppressed during the trough and suppressed during the peak (*Figure 10B*, bottom left). The opposite was the case for gamma. It barely affected or moderately suppressed distal inputs (*Figure 10B*, top right), while proximal inputs were enhanced during the trough and suppressed during the peak (*Figure 10B*, bottom right).

Summarizing these results (*Figure 10C*), we found that somatic spiking driven by clustered proximal synapses was bidirectionally modulated by gamma rhythms and suppressed by beta. On the other hand, spiking driven by distal clusters was bidirectionally modulated by the beta rhythm and suppressed by gamma. Thus, both rhythms regulate the sensitivity of pyramidal neurons to afferents throughout the dendritic tree, but in a counterposed location-dependent manner.

## Discussion

Arising from multiple interneuron subtypes, inhibition sculpts pyramidal neuron activity by acting at different membrane regions and distinct rhythmic frequencies (*Figure 11A*). Little was known about how these factors interact with the complexity of dendritic integration. To address this, we characterized the interaction between the location and rhythmicity of inhibition on integration in a morphologically and biophysically detailed L5 pyramidal neuron model with active dendrites. We found that distal dendritic inhibition modulated the occurrence of dendritic spikes, while perisomatic inhibition altered action potential generation. This translated into location-specific differences in the effectiveness of inhibitory rhythms. Beta rhythmic inhibition entrained dendritic spikes, focusing them into the phase when inhibition was at a minimum, but only when delivered to the distal dendrites (*Figure 11B*).

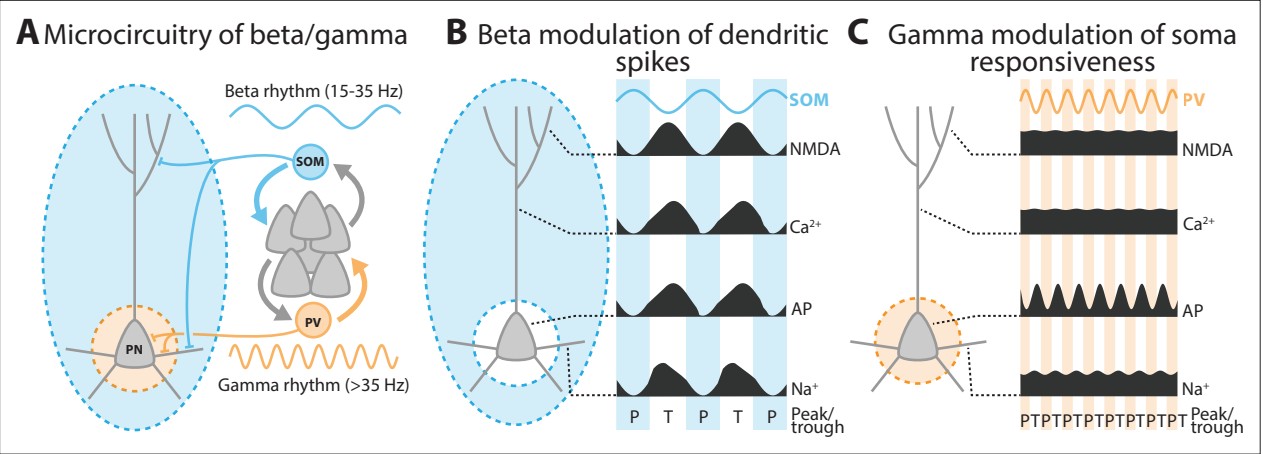

**Figure 11.** A summary schematic of the principal findings. (**A**) The microcircuitry that was simulated in this study. (**B**) Beta rhythmic inhibition to the distal dendrites modulated dendritic spikes. (**C**) Gamma rhythmic inhibition to the perisomatic region modulated action potential initiation. AP stands for action potentials.

In contrast, gamma modulated the threshold for action potential initiation, but only when delivered perisomatically (*Figure 11C*). The effects of these rhythms were frequency specific, with the timing of beta and gamma aligning preferentially with phase-dependent effects on neuronal integration. As a likely result, beta oscillations bidirectionally controlled transmission in distal dendrites and suppressed those onto proximal dendrites, while gamma oscillations did the opposite.

This indicates that the spatial targeting and timing of inhibition go hand in hand. Their alignment is thus fundamental to their distinct effects on neuronal integration. And in turn, this affects function at multiple levels, from synaptic responsiveness to neural coding and microcircuit operation.

## Neuronal responsiveness

Oscillatory rhythms potentially affect synaptic integration through several mechanisms. Early theorizing held that oscillations synchronize populations of excitatory neurons such that their efferent synapses benefit from spatiotemporal summation in eliciting action potentials (*Salinas and Sejnowski, 2000*). Later, it was proposed that if neurons in two regions exhibited coherent variations in their somatic membrane potentials, bringing them closer to or farther from action potential threshold, they could preferentially exchange activities (*Fries, 2005*). Modeling and in vitro studies have shown that the phase of a sinusoidal current injected at the soma modulates the timing of evoked action potentials (*Hopfield, 1995*) or their probability of occurrence (*Volgushev et al., 1998*). Similar gamma phase dependence has been found in vivo (*Vinck et al., 2010a*).

Rhythmic inhibition may also periodically modulate the sensitivity of the neuron to synaptic input. Inhibitory synapses achieve this by lowering the membrane resistance, i.e., shunting (*Prescott and De Koninck, 2003*). This makes it harder for an excitatory synaptic current to drive the membrane voltage toward the action potential threshold. Indeed, optogenetically stimulating PV interneurons at gamma frequencies modulates the responsiveness of cortical neurons to sensory inputs in a phase-dependent manner (*Cardin et al., 2009*). However, normally PV interneurons are driven by local principal cells, so the inhibition they impose reflects the aggregate activity in the local network (leading to E/I balance; *Ferguson and Gao, 2018*). Model networks wired in this manner emit spontaneous gamma bursts that produce phase-dependent modulation of the relationship between EPSP amplitude and spiking probability, with a strong positive relationship during the trough of gamma and a weak one during the peak (*Feng et al., 2019*). Consistent with these results, putative monosynaptically connected pairs of single units recorded in vivo show an increase in spike transmission probability during the trough of gamma (*Headley et al., 2021*).

To summarize the present possibilities, oscillations could modulate a neuron's responsiveness by (1) synchronizing the synaptic inputs impinging upon it, (2) modulating how close the membrane potential is to the action potential threshold, or (3) varying its sensitivity to synaptic inputs via shunting inhibition.

Our results fill out this picture in two ways. First, we add a fourth mechanism, which is (4) the modulation of dendritic spiking events. Second, we found that beta and gamma oscillations differentially engaged these mechanisms. Beta oscillations were primarily caused by 2 via 4 and affected 3. Gamma, on the other hand, operated primarily through 3. It should be noted that to isolate these effects, our model had Poisson excitatory synaptic inputs, which precluded mechanism 1, afferent synchrony. However, it is likely the degree of synchrony would impact both rhythms. Supporting this, in vivo whole-cell recordings from neurons in mouse visual cortex exhibit brief membrane depolarizations in synchrony with gamma (*Perrenoud et al., 2016*).

## Relevance to coding

Given that beta and gamma rhythms influence spiking via distinct mechanisms, they may also differentially impact coding. During gamma bursts in V1, neurons show a stronger modulation of their firing rate by the contrast of grating stimuli (*Perrenoud et al., 2022*). Gamma phase also modulates orientation selectivity and correlated variability between neurons (*Womelsdorf et al., 2012*). This enhancement is modulated by attention (*Lisitsyn et al., 2020*). Our results indicate that these effects reflect shifts in somatic sensitivity to excitatory synaptic inputs.

Much less is known about the influence of beta oscillations on coding. In the visual cortex of mice, activation of SOM interneurons increased with the visual stimulus size and homogeneity (*Veit et al., 2017*; note that this study referred to the rhythm as low gamma). In a different study, using a sequence working memory task in primates, the order of presented items corresponded to their phase in the beta rhythm (*Siegel et al., 2009*). However, this does not indicate how coding varies with beta phase. Considering our finding of beta-synchronized dendritic spikes, it may operate by enabling the summation of normally asynchronous afferents on distal dendrites, thereby facilitating their ability to drive somatic spiking.

One interpretation of rhythms arising from local inhibitory feedback is that they maintain the balance between excitation and inhibition. This can be thought of as a normalization operation that maintains activity within a set range. Normalization can be achieved either through a subtractive effect that raises the threshold for initiating an action potential or a multiplicative effect that lowers the slope of the relationship between excitation and action potential firing rate. When considered at the population level, these normalization effects impact coding in different ways. Subtractive normalization increases sparsity by dropping out neurons whose excitation is below the raised threshold. Multiplicative normalization, however, encourages dense codes by scaling down firing rates and compressing the range of firing rates. This study found that while both perisomatic and distal dendritic inhibition produced subtractive effects, only perisomatic had a multiplicative effect. Tying this to beta and gamma, beta rhythms may encourage sparse population codes, while gamma allows for dense.

## Interaction with microcircuitry

Pyramidal neurons are embedded in a cortical column where they are sparsely connected among themselves and densely connected with local interneurons. The dense connectivity with interneurons is crucial for the generation of local beta and gamma rhythms (*Chen et al., 2017*; *Veit et al., 2017*). How might this circuitry interact with the differential regulation of dendritic integration by beta and gamma?

We found that beta rhythms preferentially modulated a pyramidal neuron's response to inputs on its distal dendrites. The apical tuft in the superficial layers of cortex primarily receives long-range inputs from regions higher up in the cortical hierarchy (*D'Souza and Burkhalter, 2017*; *Harris et al., 2019*). This makes beta rhythms ideally positioned to modulate these top-down/feedback signals. Since the rhythm likely derives from local activation of SOM interneurons, it is worth considering what situations would activate those and how they might relate to the functioning of apical dendrites. SOM interneurons receive facilitating synapses from local pyramidal neurons (*Beierlein et al., 2003*; *Campagnola et al., 2022*), which would make them especially sensitive to bursts of action potentials arising from $Ca^{2+}$ spikes at the apical nexus (*Larkum et al., 2009*). Since $Ca^{2+}$ spikes are driven by synaptic activation in the apical tuft, it is likely that beta rhythms regulate the generation of action potential bursts arising from long-range inputs.

Particularly in the visual cortex, SOM interneurons can generate a rhythm in the 25–30 Hz range (*Veit et al., 2017*). We found this to be at the upper end of the frequency range for dendritic inhibitory

rhythms to be effective in modulating NMDA and $Ca^{2+}$ spikes. If this rhythm solely recruited SOM interneurons, its effectiveness would be marginal. Potentially compensating for this, recent work has found that PV interneurons also participate in beta/low gamma (*Onorato et al., 2025*; *Tahvili et al., 2025*) (but see *Chen et al., 2017*; *Veit et al., 2017*). In our model, on its own, when beta rhythmic inhibition was delivered perisomatically, we found that it was less able to entrain spiking and had an overall hyperpolarizing effect. However, if delivered in conjunction with the distal dendritic inhibition arising from SOM interneurons, this may strengthen entrainment.

Turning to gamma, our results indicated that it mainly affects processing in the soma and proximal dendrites. These are preferentially targeted by L5 pyramidal neurons, which sparsely interconnect with projection-type specificity (*Kawaguchi, 2017*; *Morishima et al., 2011*; *Morishima and Kawaguchi, 2006*). A confluence of evidence implicates PV interneurons in the production of high-frequency inhibitory rhythms such as gamma (*Buzsáki and Wang, 2012*; *Fernandez-Ruiz et al., 2023*). Given that PV interneurons diffusely interconnect with local pyramidal neurons (*Packer and Yuste, 2011*; *Morishima et al., 2017*), and the synapses they receive from them are depressing (*Beierlein et al., 2003*; *Campagnola et al., 2022*), they likely regulate abrupt increases in local ensemble activity. Interactions within ensembles that precede activating the PV population will be boosted, while those following will be attenuated.

## Interneuron specializations and rhythm timescales

Lastly, it is worth considering why beta and gamma rhythms are primarily mediated by different types of interneurons (but see *Tahvili et al., 2025*). One possibility is that the differing pace merely arises from the electrotonic timescales of their feedback inhibition; SOM cells target distal dendrites and thus produce a slower inhibitory feedback signal, versus PV cells delivering rapid perisomatic inhibition. Indeed, we found that swapping the cellular compartments receiving beta and gamma rhythmic inhibition impaired their effectiveness and lowered overall excitability.

So, while our results suggest that spatial targeting of SOM and PV interneurons aligns with the timescales of their *network-level* rhythms, it could also be that their timing and subcellular localization interact to produce specialized *neuron-level* functions (*Lovett-Barron et al., 2012*). For instance, NMDA and $Ca^{2+}$ spikes in the distal dendrites last for ~50 ms, making the slower beta rhythm more appropriate for bidirectionally controlling them. Both can be described as dynamical systems with distinct phases with differing sensitivity to inhibition. $Ca^{2+}$ spikes are dynamical events comprised of an initiation, plateau, and termination phase. Inhibition delivered during the plateau phase shortens their duration (*Dudai et al., 2022*). If the beta rhythm is comprised of cycling between periods of elevated excitation (increased NMDA spike generation) followed by elevated inhibition, then $Ca^{2+}$ spike initiation will tend to occur during the excitatory phase and its plateau during the subsequent inhibitory phase. A plateau during the inhibitory phase will more quickly enter termination. This is bidirectional control. On the other hand, slower rhythms (e.g. 1 Hz) initiate $Ca^{2+}$ spikes during the excitatory phase that plateau and enter termination autonomously, before the inhibitory phase is reached. The same principle holds for NMDA spikes (*Doron et al., 2017*). As a result, rhythms in the range from 15 to 30 Hz are optimal for synchronizing the onsets and offsets of dendritic spikes across a population of neurons.

The integrative effects of gamma (>40 Hz) are also specialized. Low-frequency inhibitory rhythms delivered to the soma tended to shift the membrane potential higher or lower with the rhythm's phase, effectively bringing it closer or farther from action potential threshold but not changing the neuron's sensitivity to fast synaptic inputs. In the gamma frequency range, this is reversed, with the mean membrane potential not varying with rhythm phase but with a shifting bias to positive or negative membrane potential fluctuations. In addition, the trough phase of gamma lowers the threshold for action potential initiation, while slower rhythms like beta only raise the threshold. Consequently, the timing of gamma is ideal for increasing the sensitivity of the neuron to rapid excitation. This agrees with the observation that gamma oscillations accompany rapid excitation-inhibition balancing (*Atallah and Scanziani, 2009*).

## Conclusion

For the most part, the study of beta and gamma rhythms has focused on their correlation with task-related events and entrainment of neuronal firing. Fundamentally, these effects derive from their

influence on neuronal integration, which has multiple stages, from modulating the postsynaptic response, inducing a dendritic spike, and triggering an action potential. Since beta and gamma rhythms differentially impacted these processes, this invites a reappraisal of their role in coding and communication, along with unveiling a new hypothesis space for understanding their function.

## Methods

We adapted a previously published model of an L5 pyramidal neuron (*Egger et al., 2020*; *Hay et al., 2011*) to the Brain Modeling Toolkit (*Dai et al., 2020*) format to facilitate reproducibility. This open-source software utilized NEURON 7.7 (*Carnevale and Hines, 2009*) with an integration time step of 0.1 ms for simulating the membrane potential. We added synapses and designed inputs in accordance with the literature to study synaptic integration in vivo. Details of the passive membrane properties and active conductances are only described briefly (full details can be found in *Hay et al., 2011*). Our focus here is on describing the enhancements we incorporated to mimic in vivo features in the model neuron.

### L5 pyramidal cell compartmental model

Briefly, the cell was built on a morphological reconstruction of an L5 pyramidal tract neuron and included 10 active Hodgkin-Huxley conductances with kinetics taken from the literature (*Kawaguchi and Kubota, 1997*). These channels were a fast-inactivating $Na^+$ current ($I_{NaT}$), persistent $Na^+$ current ($I_{NaP}$), nonspecific cation current ($I_h$), muscarinic $K^+$ current ($I_m$), slow inactivating $K^+$ current ($I_{Kp}$), fast inactivating $K^+$ current ($I_{Kt}$), fast non-inactivating $K^+$ current ($I_{Kv3.1}$), high voltage-activated $Ca^{2+}$ current ($I_{Ca\_HVA}$), low voltage-activated $Ca^{2+}$ current ($I_{Ca\_LVA}$), and $Ca^{2+}$ activated $K^+$ current ($I_{SK}$). The somatic compartments contained all of these channels except $I_m$. Both the apical and basal dendrite compartments contained $I_{Ca\_LVA}$, $I_{Ca\_HVA}$, $I_{SK}$, $I_{Kv3.1}$, $I_{NaT}$, $I_m$, and $I_h$. The membrane capacitance was set to 1 $\mu F/cm^2$ for the soma and axon and 2 $\mu F/cm^2$ for the basal and apical dendrites to correct for dendritic spine area. Axial resistance was set to 100 W-cm, and leak reversal potential was –90 mV.

### Divergence, probability of transmission, and functional clustering

Experimental studies indicate that each presynaptic excitatory neuron forms 2–8 synapses with its pyramidal partners, with a release probability between 0.16 and 0.9 (*Branco and Staras, 2009*). Further, the 2–8 synaptic contacts occur over a span of 20–100 µm (*Markram et al., 1997*). We constrained our model to have these properties. In addition, since synapses may cluster on dendritic branches such that functional presynaptic cell assemblies activate the same branch within 5–10 µm (*Chen et al., 2011*; *Takahashi et al., 2012*), we forced synapses coming from the same presynaptic cell assembly to cluster in this way. In summary, a single presynaptic input fiber could have 2–8 synapses, drawn from a uniform distribution with a release probability between 0.16 and 0.9, also drawn from a uniform distribution (*Branco and Staras, 2009*). And the synapses from a functional group formed clusters within 5–10 µm on the same dendritic branch over a span of 100 µm.

### Synaptic conductances

We randomly distributed excitatory synaptic conductances following previous modeling work (*Goetz et al., 2021*). Maximum conductance values followed a log-normal distribution with a mean of 0.2 nS and standard deviation of 0.345 nS for excitatory synaptic conductances. Inhibitory synaptic conductances were fixed at 1 nS (*Egger et al., 2020*). We validated these conductances against excitatory and inhibitory postsynaptic currents (PSCs) measured in vitro. The model's values are compared with experimental values in *Table 1*.

### Synapse density

Synapse density was based on a comprehensive report (*Karimi et al., 2020*) that not only quantified the number of spines but also counted the synapses by identifying vesicle clouds and postsynaptic density using serial block-face electron microscopy. The authors found the apical tuft synapse density per micron as 2.16±0.16 for excitatory and 0.22±0.02 for inhibitory synapses (mean ± standard error of the mean). Other studies that only quantified spine density may undercount the true number of synapses by about three times because one can only count lateral spines, not those in the z-axis (*Horner and Arbuthnott,*

**Table 1.** Inputs to layer 5 (L5) PN.

L5 PN dendrites can course up to L1 and receive both excitatory and inhibitory inputs to their dendrites. Here, we quantified, where possible, the experimental values for synaptic magnitude, firing rate, divergence, and release probability. We matched the model parameters to the experimental values as closely as possible while preserving a reasonable basal firing rate.

| Synapse type | Characteristics | Model | Experimental |
|---|---|---|---|
| Excitatory (basal) | Magnitude of EPSCs | 37.0±32.3 pA | 30.6±29.9 pA *Morishima et al., 2011* |
| | Firing rate | 4.43±2.9 Hz | 4.43±2.9 Hz (Headley personal comm.) |
| | Divergence | 2–8 | 2–8 (*Markram et al., 1997*; *Reimann et al., 2015*; *Deuchars et al., 1994*) |
| | Number of synapses | 10042 | 10042 (*Karimi et al., 2020*) |
| | Release probability | 0.53±0.22 | 0.53±0.22 (*Brémaud et al., 2007*) |
| Excitatory (apical) | Magnitude | 25.9±24.9 pA | 30.6±29.9 pA (*Morishima et al., 2011*) |
| | Firing rate | 4.43±2.9 Hz | 4.43±2.9 Hz (Headley personal comm.) |
| | Divergence | 2–8 | 2–8 (*Markram et al., 1997*; *Reimann et al., 2015*; *Deuchars et al., 1994*) |
| | Number of synapses | 16070 | 16070 (*Karimi et al., 2020*) |
| | Release probability | 0.53±0.22 | 0.53±0.22 (*Brémaud et al., 2007*) |
| Inhibitory (perisomatic and somatic) | Magnitude | 162.5±103.1 pA | 208.3±58.7 pA (*Xiang et al., 2002*) |
| | Firing rate | 16.9±14.3 Hz | 16.9±14.3 Hz (*Yu et al., 2019*) |
| | Divergence | 2.8±1.9 | 2.8±1.9 |
| | Number of synapses | 406 | 406 (*Karimi et al., 2020*) |
| | Release probability | 0.88±0.05 | 0.88±0.05 (*Xiang et al., 2002*) |
| Inhibitory (basal) | Magnitude | 24.3±18.4 pA | 26.5±1.6 pA (*Xiang et al., 2002*) |
| | Firing rate | 3.9±4.9 Hz | 3.9±4.9 Hz (*Yu et al., 2019*) |
| | Divergence | 2.7±1.6 | 2.7±1.6 (*Tanaka et al., 2011*; *Thomson et al., 1996*) |
| | Number of synapses | 1023 | 1023 (*Karimi et al., 2020*; *Jadi et al., 2012*) |
| | Release probability | 0.72±0.10 | 0.72±0.10 (*Xiang et al., 2002*) |
| Inhibitory (apical) | Magnitude | 24.3±33.1 pA | 26.5±1.6 pA (*Xiang et al., 2002*) |
| | Firing rate | 3.9±4.9 Hz | 3.9±4.9 Hz (*Yu et al., 2019*) |
| | Divergence | 12±3 | 12±3 (*Silberberg and Markram, 2007*; *Vezoli et al., 2021*) |
| | Number of synapses | 1637 | 1637 (*Karimi et al., 2020*) |
| | Release probability | 0.30±0.08 | 0.30±0.08 (*Silberberg and Markram, 2007*) |

*1991*). With that adjustment, the other values reported – 0.47±0.01 spines/μm (*Tjia et al., 2017*) and 0.53±0.05 spines/μm (*Davidson et al., 2020*) – are roughly in alignment with that in *Karimi et al., 2020*. We used the values from *Karimi et al., 2020*, for excitatory synapse density because we found the higher value for excitatory synapse density was needed to achieve a reasonable baseline firing rate.

Morphological measurements (length of dendritic cable in μm) combined with these synapse density estimates yield the following estimates for total number of synapses: Excitatory: 16070 (2.16 syns/μm * 7440 μm) on apical dendrites and 10042 (2.16 syns/μm * 4649 μm) on basal dendrites. Inhibitory: 150 on soma (*Egger et al., 2020*; *Markram et al., 2015*; *Reimann et al., 2015*) + (0.22 syns/μm *483 mm)=256 perisomatic, 1637 (0.22 syns/μm * 7440 μm) on apical dendrites, 1023 (0.22 syns/μm * 4649 μm) on basal dendrites.

## Synaptic kinetics and short-term plasticity

All excitatory transmission was mediated by AMPA/NMDA receptors and inhibitory transmission by GABA$_A$ receptors as in our prior models (see *Table 2*). The rise/decay constants for the excitatory

**Table 2.** Short-term presynaptic plasticity.
Parameters were tuned to match experimental recordings reported in *Campagnola et al., 2022*.

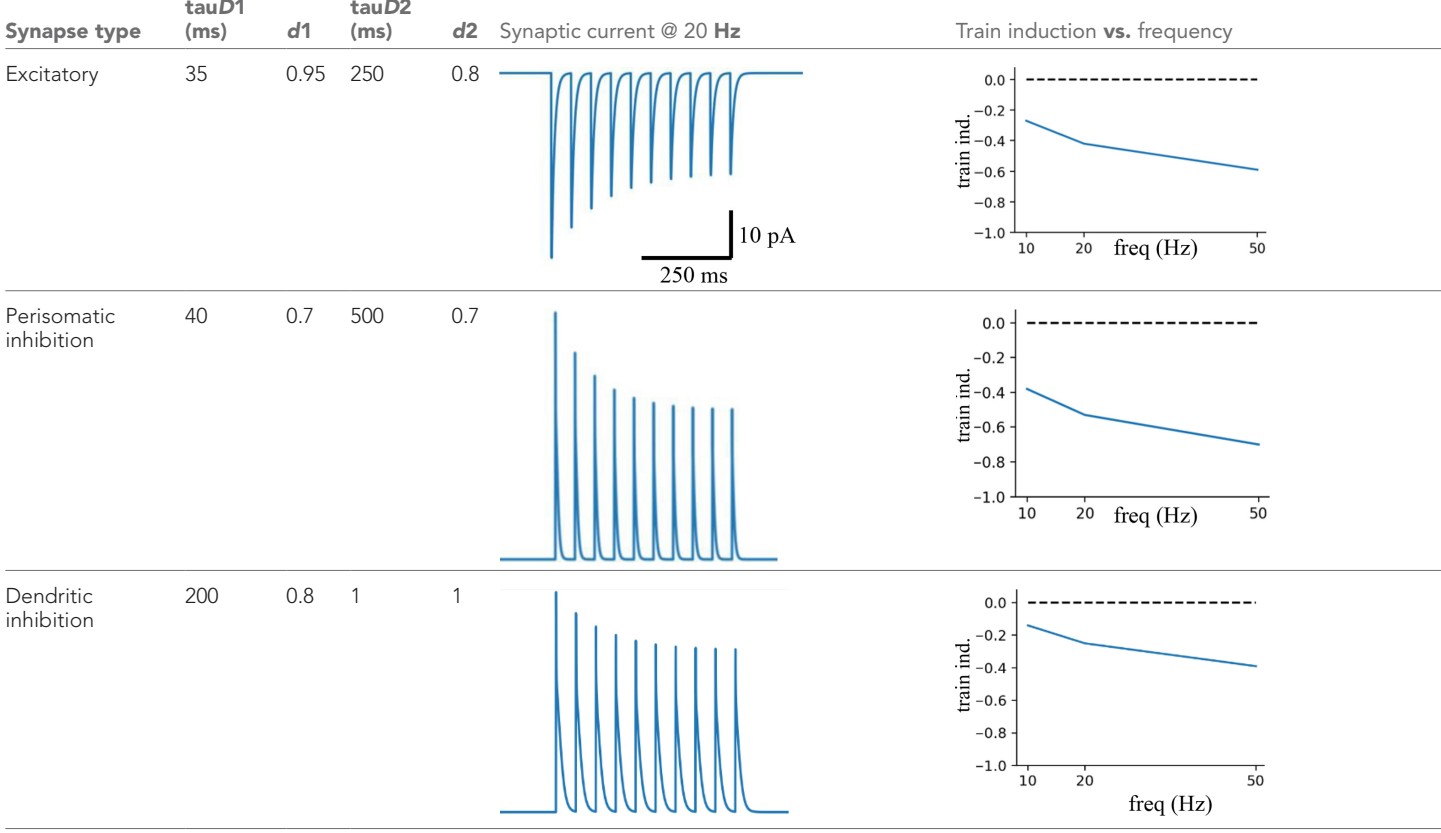

| Synapse type | tauD1 (ms) | d1 | tauD2 (ms) | d2 | Synaptic current @ 20 **Hz** | Train induction **vs.** frequency |
|---|---|---|---|---|---|---|
| Excitatory | 35 | 0.95 | 250 | 0.8 | | |
| Perisomatic inhibition | 40 | 0.7 | 500 | 0.7 | | |
| Dendritic inhibition | 200 | 0.8 | 1 | 1 | | |

connections were as follows: Rise/decay time constants of 0.6/6.9 ms for AMPA, and 3.7/125 ms for NMDA synapses, and a reversal potential of 0 for both. And for the inhibitory GABA-ergic connections, the rise/decay time constants were 0.5 and 6.8 ms, respectively, with a reversal potential of –75 mV. We implemented short-term depression in both the excitatory and inhibitory synapses (*Campagnola et al., 2022*) via an exponential decay of synaptic weight over short timescales (*Varela et al., 1997*). Following *Campagnola et al., 2022*, we calculated a short-term plasticity (STP) train-induction value by applying 8 pulses at 50 Hz to the synapse while in voltage clamp and measured the resulting PSC. The 6th, 7th, and 8th PSCs were averaged, and the 1st PSC was subtracted from the result. This value was divided by the max PSC current to yield the STP train-induction value. Excitatory synapses to L5 pyramidal neurons had an STP train-induction value of –0.34. Inhibitory synapses arising from PV+ basket interneurons making perisomatic contact had a value of –0.60 and inhibitory synapses arising from SOM+ interneurons making dendritic contact had a value of –0.10. The equations to model the synaptic currents and short-term depression are provided in the next section.

## Modeling of synaptic currents

The corresponding synaptic currents were modeled by dual exponential functions (*Destexhe et al., 1994*; *Durstewitz et al., 2000*). All excitatory transmission was mediated by AMPA/NMDA receptors and inhibitory transmission by GABA$_A$ receptors. The corresponding synaptic currents were modeled by dual exponential functions (*Destexhe et al., 1994*; *Durstewitz et al., 2000*), as shown in *Equations 1–3*:

$$I_{AMPA} = w \times G_{AMPA} \times (V - E_{AMPA})$$

$$G_{AMPA} = g_{AMPA,max} \times D_{AMPA} \times r_{AMPA} \tag{1}$$

$$r_{AMPA}' = \alpha Tmax_{AMPA} \times ON_{AMPA} \times (1 - r_{AMPA}) - \beta_{AMPA} \times r_{AMPA}$$

$$I_{NMDA} = w \times G_{NMDA} \times (V - E_{NMDA})$$

$$G_{NMDA} = g_{NMDA,max} \times D_{NMDA} \times s(V) \times r_{NMDA} \qquad (2)$$

$$r_{NMDA}' = \alpha Tmax_{NMDA} \times ON_{NMDA} \times (1 - r_{NMDA}) - \beta_{NMDA} \times r_{NMDA}$$

$$I_{GABAa} = w \times G_{GABAa} \times (V - E_{GABAa})$$

$$G_{GABAa} = g_{GABAa,max} \times D_{GABAa} \times r_{GABAa} \qquad (3)$$

$$r_{GABAa}' = \alpha Tmax_{GABAa} \times ON_{GABAa} \times (1 - r_{GABAa}) - \beta_{GABAa} \times r_{GABAa}$$

where $V$ is the membrane potential (mV) of the compartment (dendrite or soma), where the synapse is located, $I$ is the current injected into the compartment (nA), $G$ is the synaptic conductance (µS), $w$ is the synaptic weight (unitless), and $E$ is the reversal potential of the synapse (mV). $g_{x,max}$ is the maximal conductance (µS), $D$ implements STP as defined below, and $r_x$ determines the synaptic current rise and decay time constants based on the terms $\alpha T_{max}$ and $\beta$ (*Destexhe et al., 1994*). The voltage-dependent variable $s(V)$ which implements the $Mg^{2+}$ block was defined as: $s(V) = [1+0.33 \exp(-0.06\ V)]^{-1}$ (*Zador et al., 1990*). The terms $ON_{NMDA}$ and $ON_{AMPA}$ are set to 1 if the corresponding receptor is open, else to 0. To calculate the factor $D$ for depression: $\tau\_D^* dD/dt = 1 - D$ and $D$ constrained to be $\leq 1$. After each stimulus, $D$ was multiplied by a constant $d$ ($\leq 1$) representing the amount of depression per presynaptic action potential and updated as $D \rightarrow D^*d$. Between stimuli, $D$ recovered exponentially back toward 1. We modeled depression using two factors $d1$ and $d2$, with $d1$ being fast and $d2$ being slow subtypes, and $d=d1^*d2$ was constrained to be $\geq 1$.

## Synapse distribution and extrinsic inputs

We distributed 26,112 excitatory synapses over the cell. With an average of 5 synapses per presynaptic neuron, this yielded 5222 presynaptic nodes. To represent presynaptic cell assemblies, we defined 'functional groups' of presynaptic nodes. We put 100 nodes in each functional group, yielding 52 functional groups. We found that varying the number of neurons in the cell assembly did not qualitatively affect our results. To have exactly 100 nodes per functional group, we rounded the total number of nodes down to 5200. Inhibitory inputs were not defined by functional groups but followed the literature cited in *Table 1*, with more than 1 synapse per presynaptic node. For example, with 1637 inhibitory synapses on the apical dendrites and an average of 12 synapses per presynaptic node, there were about 136 inhibitory nodes that contacted the apical dendrites. The numbers and distributions of model synapses along the dendrites are shown in *Table 1*.

The extrinsic inputs to the model were represented by spike trains arriving at the corresponding synapses along the dendritic tree. The L5 pyramidal model is spatially divided into the following subdivisions: somatic, perisomatic, and proximal and distal components of the apical and basal trees. We used standard definitions for these subdivisions except for perisomatic, which is typically defined as the dendritic region within 100 µm of the soma (*Freund and Katona, 2007*).

### Tonic inputs (*Figures 1–3*)

The firing rate of each excitatory spike train was randomly sampled from an experimentally recorded distribution (Headley, personal communication). To reproduce the 1/$f$ spectrum that is widely found in neuronal spike trains, the 100 spike trains in each functional group were modulated by a unique normalized (from 0.5 to 1.5) firing rate time series that had a 1/$f$ spectrum (more slow modulation of firing than fast) that was created by filtering white noise using standard techniques. The normalized firing rate time series was then multiplied with the mean firing rate of each of the 100 presynaptic nodes. Spike trains are then generated by sampling from a Poisson distribution using the firing rate time series of each presynaptic node of the functional group. In this way, spike trains that received the same modulation trace would be statistically correlated, representing functional presynaptic cell assemblies. Since feed-forward inhibition follows excitation in time, we averaged the firing rate time series used to design the excitatory spike trains, shifted it forward in time by 4 ms and used this to modulate the firing rate of the inhibitory spike trains.

### Enhanced tonic inhibition (*Figure 4*)

Inhibitory synapses were divided into two groups, those targeting perisomatic region that originate mainly from PV+ interneurons, and those on more distal dendritic regions arising mainly from SOM+ interneurons (*Tremblay et al., 2016*; *Kubota, 2014*). Perisomatic synapses were defined as those within 100 μm of the soma with the properties specified in *Table 1* under 'Inhibitory (perisomatic and somatic)'. Distal inhibitory synapses were greater than 100 μm from the soma and had their properties specified in sections 'Inhibitory (basal)' and 'Inhibitory (apical)' of the table. To investigate if these inhibitory sources differentially regulated action potential initiation and dendritic spiking, we doubled the rate of presynaptic events to one of these inhibitory populations.

### Investigation of E/I lag (*Figure 4F–I*)

We used the same scheme to design the extrinsic inputs as in *Figures 1–3*. However, to investigate the effect of E/I lag, we increased the variance of the overall excitatory firing rate. So, while the mean rate stayed the same, we raised the modulatory trace to the fourth power so that large deviations from the mean were exaggerated. This excitatory firing rate trace was used to modulate the inhibitory firing rates (same as for *Figures 1–3*). The perisomatic and distal traces were shifted independently. One was fixed at 4 ms while the other was shifted by 4, 125, 250, or 500 ms.

### Rhythmic inhibition (*Figures 5 and 6*)

Gamma rhythms depend on PV+ interneurons, while beta rhythms arise from SOM+ interneurons (*Chen et al., 2017*). Thus, to emulate an inhibitory rhythm, we rhythmically modulated inhibitory synapses in the respective dendritic region (see description above in 'Enhanced tonic inhibition') at the appropriate frequency. Beta rhythms were emulated by modulating distal inhibitory synapses at 16 Hz with a depth of 20%. Gamma rhythms were emulated by modulating perisomatic synapses at 64 Hz with a depth of 40%. The different depths of modulation ensured that these rhythms produced similar entrainment of action potentials.

To generate inhibitory spike trains that produce rhythmic inhibition during oscillations, we modulated their collective firing rate with a sine wave:

$$r\left[t\right] = A * \left(sin\left(2 * \pi * f * t\right) + \phi\right) + \textit{off} \tag{4}$$

where $A = \frac{\textit{off}}{\frac{1}{d}-1}$, $f$ is the frequency, $t$ is a vector representing time, $\phi$ is the phase, $\textit{off}$ is the offset firing rate of the spike train being modulated, and $0 < d < 1$ is the depth of modulation which represents the amplitude of the sine wave relative to $\textit{off}$. To generate the spike train, a random vector $x[t]$ was created with values uniformly distributed between 0 and 1. A spike was generated if $x\left[t\right] \leq r\left[t\right] dt$, where d$t$ in our case was 1/1000.

### Varying frequency of rhythmic inhibition (*Figures 7 and 8*)

We reran the simulations used in *Figures 5 and 6*, but across a range of frequencies that extended above and below the beta and gamma frequencies we used in all other simulations (0.5, 2, 5, 10, 16, 20, 30, 40, 50, 60, 80 Hz).

### Bursty inhibition (*Figure 9*)

We applied an envelope to the rhythmic inhibitory modulation trace to simulate the bursty temporal nature of rhythms in vivo. The underlying sine wave was produced in the same way as in the previous section. The envelope was a Gaussian kernel with a standard deviation of two oscillatory cycles.

### Clustered synaptic input (*Figure 10*)

For this experiment, we kept everything the same as in the rhythmic inhibition simulations, but introduced additional synapses driven by 40 presynaptic spike trains. These spike trains were created by choosing one Poisson process and jittering the spike times by ± 2 ms. Each spike train drove an average of 4.6 synapses for a total of 187 synapses. These were placed either on the apical dendrites near the nexus or the basal dendrites near the soma. For both nexus and basal clustered synapses, we targeted 140 μm of dendritic cable length with 187 synapses yielding a synaptic density of 1.3

synapses per micron. For the nexus case, we targeted the 140 µm region starting at the bifurcation going toward the apical dendrites. For the basal case, we targeted three basal dendrites that were directly connected to the soma. Each one had a length of 40–50 µm adding up to 140 µm total (*Figure 7A*). The synaptic weights were drawn from the distribution in *Table 1*.

### Detection of action potentials

Action potentials were detected by the NEURON software as being any time point when the somatic membrane potential crossed –10 mV with a positive first derivative.

### Detection of dendritic spikes

$Na^+$ and NMDA dendritic spike events were detected following the definitions developed in *Goetz et al., 2021*. We developed our own detection algorithm for $Ca^{2+}$ spikes since they were not previously studied. $Na^+$ spikes were detected if the voltage-gated sodium conductance density for a compartment exceeded 0.3 mS/cm². If sodium spikes were within 2 ms of a somatic action potential, they were assumed to be a consequence of a backpropagating action potential and were not counted. An example of an $Na^+$ spike can be seen in *Figure 1C1*.

NMDA spikes were detected by two criteria – voltage and current. The voltage criteria were met if the membrane potential rose above –40 mV and remained there for 26 ms or more. The current criterion was met if the NMDA current exceeded 130% of the current at the –40 mV crossing and continued to be met until the current fell to 115% of that value. Whenever both the voltage and current criteria were met, an NMDA spike was recorded. An example of an NMDA spike is shown in *Figure 1C2*.

$Ca^{2+}$ spikes were detected in the same way as NMDA spikes – using a voltage and current threshold. The only difference was instead of NMDA current for the current criteria, we used the sum of HVA and LVA calcium currents. An example is shown in *Figure 1C3*.

### Determination of electrotonic distance

We calculated electrotonic distance using NEURON's built-in impedance method. The method calculates signal attenuation between two compartments at a given frequency. We used 20 Hz because the wavelength is on a timescale comparable to the membrane time constant (10 s of milliseconds). However, changing this frequency to 10 Hz in either direction did not qualitatively affect our results.

### STA of dendritic spikes (Figures 2 and 3)

For each dendritic compartment, we created a binary time series with 2 ms bins that was true during periods where a dendritic spike was present (based on their detected start and stop times). This was done separately for each type of dendritic spike ($Na^+$, NMDA, and $Ca^{2+}$ spikes). Each then had its spike-triggered average taken around the times of somatic action potentials and their percent change from the mean across the entire simulation calculated. The STAs were then grouped together based on their electrotonic distance percentile (10% bins) from the soma and type of dendrite (apical or basal), and the median taken for each time lag.

### Generation of F-I curves (Figure 4)

The model with Poisson excitation and inhibition was used. In the *perisomatic* condition, we doubled the rate of inhibition to the somatic and proximal dendritic compartments (<100 µm from soma). In the *distal* case, we doubled the rate of inhibition to the distal dendritic compartments (>100 µm from soma). In each case, we injected 20 s current steps ranging from –1 to 2.4 nA in 200 pA intervals.

### Phase histogram of dendritic spikes (Figure 5)

Entrainment of dendritic spikes to inhibitory rhythms was calculated by first converting dendritic spikes to a binary time series as was done for the STA calculation. The inhibitory rhythm was then binned by phase, with bin widths of $\pi/4$, yielding 8 phase bins. The mean of the dendritic spike binary series was taken for each phase bin. These were converted to percent change from mean and their median taken across compartments by their electronic distance and dendritic type (the same as in the STA calculation).

## Determination of action potential threshold and somatic membrane voltage distribution (Figure 6)

Action potential threshold was taken to be the voltage 1 ms prior to the peak of the spike. Voltage distributions were calculated using 1 ms bins.

## Entrainment of dendritic spike onsets (Figure 7)

We measured the entrainment of dendritic spike onsets to the inhibitory rhythm using the pairwise phase consistency index, an unbiased metric (*Vinck et al., 2010b*).

## Phase histogram of dendritic spikes (Figure 9)

For calculating entrainment of dendritic spikes to the rhythm, we followed the same procedure as detailed above for *Figure 5*.

## CC between presynaptic spikes and action potentials (Figure 10)

We measured the effect of inhibitory rhythms on the coupling between concentrated synaptic inputs and action potentials using a phase-stratified CC. Presynaptic spike times that drove the concentrated synaptic inputs were grouped together and separated based on whether they occurred during the positive or negative phase of the imposed inhibitory rhythm. Their CC with somatic action potentials was then calculated using 1 ms bins. Since both the presynaptic and action potential times had inherent periodicities, these had to be corrected for (*Headley and Weinberger, 2013*). This was performed by converting the CC and both autocorrelations to the frequency domain using the fast Fourier transform, setting the coefficients at the inhibitory modulation rhythm to 0, performing element-wise division of the CC by the square root of the autocorrelations multiplied together, and then converting back to the time domain using the inverse fast Fourier transform on that result. To measure the strength of the CC peak, we identified the positive peak of the CC nearest to zero time lag and then integrated over its area with the edges defined by when the CC crossed from positive to negative values.

## Additional information

### Funding

| Funder | Grant reference number | Author |
|---|---|---|
| National Institutes of Health | NINDS 5R01NS123396 | Drew B Headley |
| National Institutes of Health | MH122023 | Satish S Nair |
| National Science Foundation | OAC-1730655 | Satish S Nair |
| National Science Foundation Graduate Research Fellowship Program | | Benjamin Latimer |

The funders had no role in study design, data collection and interpretation, or the decision to submit the work for publication.

### Author contributions

Drew B Headley, Conceptualization, Methodology, Writing – original draft, Writing – review and editing; Benjamin Latimer, Software, Formal analysis, Investigation, Visualization, Methodology, Writing – review and editing; Adin Aberbach, Software, Formal analysis, Investigation, Methodology; Satish S Nair, Conceptualization, Methodology, Writing – review and editing

### Author ORCIDs

Drew B Headley https://orcid.org/0000-0003-0243-3363
Satish S Nair https://orcid.org/0000-0002-1489-7029

Reviewer #1 (Public review): https://doi.org/10.7554/eLife.95562.3.sa1
Reviewer #2 (Public review): https://doi.org/10.7554/eLife.95562.3.sa2
Author response https://doi.org/10.7554/eLife.95562.3.sa3

## Additional files

### Supplementary files
MDAR checklist

### Data availability
Simulation code is deposited at ModelDB at https://modeldb.science/2019883. The raw simulation data are available from Dryad at https://doi.org/10.5061/dryad.v6wwpzhb8. Analysis code is posted as a GitHub repo at https://github.com/dbheadley/InhibOnDendComp (copy archived at *Headley, 2025*).

The following dataset was generated:

| Author(s) | Year | Dataset title | Dataset URL | Database and Identifier |
| --- | --- | --- | --- | --- |
| Drew H, Ben L, Satish N | 2026 | Data from: Spatially targeted inhibitory rhythms differentially affect neuronal integration | https://doi.org/10.5061/dryad.v6wwpzhb8 | Dryad Digital Repository, 10.5061/dryad.v6wwpzhb8 |

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
