## [Editor Report · eLife Assessment]

This **valuable** study assesses through simulations how several features of local cortical circuits - interneuron subtypes, their specific targeting of dendritic compartments, and certain brain rhythms - together affect the integration of synaptic inputs by a pyramidal cell. Employing several carefully considered simulation setups they **convincingly** demonstrate that beta rhythms are best suited to modulate and control dendritic Ca-spikes while gamma rhythms affect their coupling to somatic spiking, or how basal inputs are directly integrated into somatic spikes. However, the baseline setup may be idealized for the generation of the events in question and it would be beneficial if the similarity to the in-vivo activity regime was demonstrated further. The results will be relevant for neuroscientists studying local circuits or developing more abstract theories at the systems level.

---

## [Referee Report · Reviewer #1 (Public review)]

In this study, the authors explore the implications of two types of rhythmic inhibition - "gamma" (30-80 Hz) and "beta"(13-30Hz) - for synaptic integration. They study this in a multi-compartmental model L5 pyramidal neuron with Poisson excitation and rhythmic inhibition (16 Hz and 64 Hz), applied either to the perisomatic or apical tuft regions in the neuron. They find that 64 Hz inhibition applied to the cell body is effective in phasic modulation of AP generation, while 16 Hz inhibition applied to the apical tufts is effective in phasic modulation of dendritic spikes (in addition to APs). Switching the location of the two kinds of rhythmic inhibition reduces the overall excitability, but is not effective in phasic modulation of either dendritic spikes and weakly so for somatic APs.

Strengths:

The effect of the timescale of rhythmic inhibition on synaptic integration is an interesting question, since (a) rhythmic spiking is most strongly evident in inhibitory population, (b) rhythmic spiking is modulated by behavioral states and the sensory environment. The methods are clear and data are well-presented. The study systematically explores the effect of two frequencies of rhythmic inhibition in a biophysically detailed model. The study considers not only idealized rhythmic inhibition but also the bursty kind that is observed in in-vivo conditions. Both distributed and clustered excitatory synaptic organization are simulated, which covers the two extremes of the spatial organization of excitatory inputs in-vivo.

---

## [Referee Report · Reviewer #2 (Public review)]

Summary:

The manuscript illustrates how spatial targeting (perisomatic vs distal, apical and basal dendritic) and timing of inhibition is crucial to distinct effects on neuronal integration, and show that beta and gamma oscillations differentially engage dendritic spiking mechanisms.

Strengths:

The strength of this study lies in the integrative biophysical modelling of a layer 5 pyramidal neuron by bringing together in vitro and in vivo observations

Weaknesses:

The weaknesses are probably in some of the parameterization of inhibitory synaptic dynamics. A unitary peak conductance of 1nS is very high for inhibitory synapses. This high value could invariably skew some of the network-level predictions. The authors could obtain specific parameters from the Neocortical Collaboration Portal (https://bbp.epfl.ch/nmc-portal/microcircuit.html), which comes across an incredible resource for cortical neurons and synapses.

---

## [Author Response]

The following is the authors’ response to the original reviews.

**Reviewer #1:**
SOM+ interneurons such as Martinotti cells target the apical tufts of pyramidals in the cortex. Since interneurons in general are strongly implicated in mediating rhythmic population activity over a range of timescales, it is quite appropriate to study the consequence of rhythmic inhibition provided by SOM+ interneurons for synaptic integration, including the phenomenon of dendritic spikes. However, using conclusions from a singular study (ref 22) to identify the beta band as the rhythm mediated by SOM+ is not very accurate. SOM+ interneurons have been implicated in regulating rhythms centered just below 30 Hz (refs 22, 21). It is a range that lies in the grey zone of the traditional definition of beta and gamma. However, it is significantly higher than the 16 Hz rhythms explored in this study. It thus remains unknown how a 25-30 Hz rhythmic inhibition (that has an experimentally suggested role for dendrite targeting SOM+ INs) in apical tufts regulates dendritic spikes.

We agree with the reviewer that the rhythms arising from SOM+ interneurons can extend their frequencies higher than the 16 Hz analyzed in this study. To address this, we have conducted a new set of simulations where we delivered distal dendritic inhibition across a range of frequencies, from 0.5 to 80 Hz (see new Results section “Frequency specific effects of rhythmic inhibition on neuronal integration”). These results revealed, surprisingly, that at 30 Hz their ability to entrain Ca^2+^ and NMDA spikes degrades (but not Na^+^ spikes). This suggests that beta rhythms in the 20-30 Hz range are operating at the highest frequency for which dendritically targeting inhibition will be effective. The implications are covered in the Discussion section “Interaction with microcircuitry”. They are:

“Particularly in the visual cortex, SOM interneurons can generate a rhythm in the 25-30 Hz range [22]. We found this to be at the upper end of the frequency range for dendritic inhibitory rhythms to be effective in modulating NMDA and Ca^2+^ spikes. If this rhythm solely recruited SOM interneurons, its effectiveness would be marginal. Potentially compensating for this, recent work has found that PV interneurons also participate in beta/low-gamma [23, 24] (but see [21, 22]). In our model, on its own when beta rhythmic inhibition was delivered perisomatically we found that it was less able to entrain spiking and had an overall hyperpolarizing effect. However, if delivered in conjunction with the distal dendritic inhibition arising from SOM interneurons, this may strengthen entrainment.”

Distal dendritic inhibition has been previously shown to be more effective in controlling dendritic spikes. However, given the slow timescale of dendritic spikes, it can be hypothesized that high-frequency rhythmic inhibition would be ineffective in entraining the dendritic spikes either in distal or proximal location, as demonstrated by 4H and 5F, and vice versa. A computational study can take this further by exploring the robustness of this hypothesis. By sticking to a single-frequency definition of what constitutes Gamma (64 Hz) and Beta (16 Hz) inhibition, the current exploration does support the core hypothesis. However, given the temporal dynamics of dendritic spikes, it is valuable to learn, for example, the upper bound of "Beta" range (13-30Hz) inhibition that fails to phasically modulate them. In addition to the reason stated in the earlier paragraph, Alpha band activity (8-12 Hz), has been implicated (e.g. van Kerkoerle, 2014) in signaling of inter-areal feedback to the superficial layer in the cortex, potentially targeting apical tufts of pyramidals from multiple layers and resulting in alpha-range rhythmic inhibition. To make the findings significant, it might therefore be more pertinent to understand the consequences of ~10Hz rhythmic inhibition (in addition to the ~25-30 Hz Beta/Gamma) in the apical tufts for phasic modulation of dendritic spikes.

We added an additional set of simulations that address this in the Results section ‘Frequency specific effects of rhythmic inhibition on neuronal integration’. In general, we found that dendritic and perisomatic inhibitory rhythms at lower frequencies could entrain AP generation, but with less functional specialization. This is explored in our Discussion section ‘Interneuron specializations and rhythm timescales’.

The differential effect of Gamma and Beta range inhibition on basal and apical excitatory clusters is not convincing from the information provided. The basal cluster appears to overlap with perisomatic inhibitory synapses. The description in the methods does not have enough information to negate the visual perception (ln 979-81). With this understanding, it is not surprising that the correlation between excitation and APs is high (during the trough of gamma) for basal and not apical excitation. A more comparable scenario would be a more distal location of the basal excitatory cluster.

While we stated in the original manuscript that we were contrasting ‘basal’ vs. ‘apical’ clustered inputs, this terminology did not reflect our intent with these analyses. We meant to contrast proximal vs. distal dendritic clustered synaptic inputs, which the reviewer correctly noted is confounded in the apical vs. basal comparison. We have rewritten these results, their discussion, and corresponding figure, to clearly state that we are contrasting proximal vs. distal synaptic input.

**Reviewer #2:**
The weaknesses are probably in some of the parameterizations of inhibitory synaptic dynamics. A unitary peak conductance of 1nS is very high for inhibitory synapses. This high value could invariably skew some of the network-level predictions. The authors could obtain specific parameters from the Neocortical Collaboration Portal (https://bbp.epfl.ch/nmcportal/microcircuit.html), which is an incredible resource for cortical neurons and synapses.

We appreciate the valuable resource mentioned by the reviewer and will consult it when constructing future models. Regarding the present one, our choice of peak conductance was based on previous studies, namely:

Egger R, Narayanan RT, Guest JM, Bast A, Udvary D, Messore LF, Das S, de Kock CPJ, Oberlaender M (2020) Cortical output is gated by horizontally projecting neurons in the deep layers. Neuron 105, 122-137.e128.

and

Xiang Z, Huguenard JR, Prince DA (2002) Synaptic inhibition of pyramidal cells evoked by different interneuronal subtypes in layer v of rat visual cortex. J Neurophysiol 88, 740-750.

The study by Egger et al. used an inhibitory peak conductance of 1 nS and was simulating circuitry very similar to ours. We validated these synapses in pilot simulations that sought to characterize the resulting IPSPs and IPSCs, and whose results can be seen in Table 1 of our methods. These synapses exhibited IPSCs whose peak amplitudes ranged over values (~24162 pA) that agreed with the experimental literature, such as Xiang et al.

Given this, we feel our parameterization of inhibitory synapses does not warrant any changes.

**Reviewer #3:**
What disappointed me a bit was the lack of a concise summary of what we learned beyond the fact that beta and gamma act differently on dendritic integration. The individual paragraphs of the discussion often are 80% summary of existing theories and only a single vague statement about how the results in this study relate. I think a summarizing schematic or similar would help immensely.

We agree with the reviewer that a summary schematic would help the reader. This has been added to the manuscript as Figure 11. It demonstrates the principal findings of the paper and is referenced in the opening paragraph of the discussion section.

Orthogonal to that, there were some points where the authors could have offered more depth on specific features. For example, the authors summarized that their "results suggest that the timescales of these rhythms align with the specialized impacts of SOM and PV interneurons on neuronal integration". Here they could go deeper and try to explain why SOM impact is specialized at slower time scales. (I think their results provide enough for a speculative outlook.)

This discussion has been expanded under the section “Interneuron specializations and rhythm timescales”. The added text is:

“So, while our results suggest that spatial targeting of SOM and PV interneurons aligns with the timescales of their network-level rhythms, it could also be that their timing and subcellular localization interact to produce specialized neuron-level functions [85]. For instance, NMDA and Ca^2+^ spikes in the distal dendrites last for ~50 ms, making the slower beta rhythm more appropriate for bidirectionally controlling them. Both can be described as dynamical systems with distinct phases with differing sensitivity to inhibition. Ca^2+^ spikes are dynamical events comprised of an initiation, plateau, and termination phase. Inhibition delivered during the plateau phase shortens their duration [86]. If the beta rhythm is comprised of cycling between periods of elevated excitation (increased NMDA spike generation) followed by elevated inhibition, then Ca^2+^ spike initiation will tend to occur during the excitatory phase, and its plateau during the subsequent inhibitory phase. A plateau during the inhibitory phase will more quickly enter termination. This is bidirectional control. On the other hand, slower rhythms (e.g. 1 Hz) initiate Ca^2+^ spikes during the excitatory phase that plateau and enter termination autonomously, before the inhibitory phase is reached. The same principle holds for NMDA spikes [87]. As a result, rhythms in the range from 15-30 Hz are optimal for synchronizing the onsets and offsets of dendritic spikes across a population of neurons.

The integrative effects of gamma (>40 Hz) are also specialized. Low frequency inhibitory rhythms delivered to the soma tended to shift the membrane potential higher or lower with the rhythm’s phase, effectively bringing it closer or farther from AP generation but not changing the neuron’s sensitivity to fast synaptic inputs. In the gamma frequency range, this is reversed, with the mean membrane potential not varying with rhythm phase but with a shifting bias to positive or negative membrane potential fluctuations. In addition, the trough phase of gamma lowers the threshold for AP generation, while slower rhythms like beta only raise the threshold. Consequently, the timing of gamma is ideal for increasing the sensitivity of the neuron to rapid excitation. This agrees with the observation that gamma oscillations accompany rapid excitation-inhibition balancing [88].”

We also extended our discussion section ‘Relevance to coding’ to explore how beta and gamma rhythms can support sparse vs. dense population coding, respectively. It reads:

“One interpretation of rhythms arising from local inhibitory feedback is that they maintain the balance between excitation and inhibition. This can be thought of as a normalization operation that maintains activity within a set range. Normalization can be achieved either through a subtractive effect that raises the threshold for initiating an action potential, or a multiplicative effect that lowers the slope of the relationship between excitation and action potential firing rate. When considered at the population level, these normalization effects impact coding in different ways. Subtractive normalization increases sparsity by dropping out neurons whose excitation is below the raised threshold. Multiplicative normalization, however, encourages dense codes by scaling down firing rates and compressing the range of firing rates. This study found that while both perisomatic and distal dendritic inhibition produced subtractive effects, only perisomatic had a multiplicative effect. Tying this to beta and gamma, beta rhythms may encourage sparse population codes while gamma allows for dense.”

Beyond that, the authors invite the community to reappraise the role of gamma and beta in coding. This idea seems to be hindered by the fact that I cannot find a mention of a release of the model used in this work. The base pyramidal cell model is of course available from the original study, but it would be helpful for follow-up work to release the complete setup including excitatory and inhibitory synapses and their activation in the different simulation paradigms used. As well as code related to that.

We have added a Code and Data Availability section that addresses this. It reads: “Simulation code is deposited at ModelDB at https://modeldb.science/2019883 . The raw simulation data are available from DBH upon request. Analysis code is posted as a github repo at https://github.com/dbheadley/InhibOnDendComp.”